# RADIOLOGIST-LIKE PROGRESSIVE RADIOLOGY RE-PORT GENERATION AND BENCHMARKING

## ABSTRACT

Radiology report generation is a critical application at the intersection of radiol-ogy and artificial intelligence. It aims to reduce radiologists' workload by au-tomating the interpretation and reporting of medical images. Previous works have employed diverse approaches, with some focusing solely on imaging data while others incorporate the indication but often neglect the interrelationships among different report sections. Our work identifies and harnesses the intrinsic relation-ships between the *indication*, *findings*, and *impression* sections of a radiology re-port. The indication section provides the clinical context and specifies the reason for the examination, setting the stage for targeted image analysis. The findings section details the radiologist's observations from the image, including identified abnormalities and relevant normal findings. The impression section synthesizes these observations to form a diagnostic conclusion, directly addressing the clinical query presented in the indication. By mapping these relationships, we propose a **R**adiologist-**L**ike **P**rogressive **G**eneration (RLPG) framework that mirrors the ra-diologist's workflow for report generation. Initially, an image encoder and a large language model process the imaging data alongside the indication to generate de-tailed findings. Subsequently, the same image, the indication, and the predicted findings are utilized to produce a concise impression. This method improves the alignment between report sections and improves the clinical relevance of the gen-erated reports. To facilitate research and benchmarking in report generation, we introduce MIMIC-1V3 (*i.e.*, 1 case vs. 3 sections), a curated dataset derived from the MIMIC-CXR by dividing each report into three sections: indication, findings, and impression. The new dataset, in conjunction with our progressive framework design, fosters advancements in automated report generation by providing a more accurate and clinically relevant solution.

## 1 INTRODUCTION

In real-world medical practice, creating a radiology report (see Fig. 1 (a)) starts with an indication from the ordering physician. The indication provides rich clinical context, often containing a clinical question (*i.e.*, the reason for the examination) and the patient's brief medical history. Subsequently, the radiologist interprets the imaging study within the clinical context. The radiology report rep-resents the sum of a radiologist's insight into the patient's condition. It mainly has two sections: findings and impression. The findings section provides an accurate radiologic description of all ab-normalities with pertinent negatives. The *impression* answers the clinical question and reflects the meaning of findings, leading to a diagnosis (Hartung et al., 2020). However, radiologist workload has increased significantly within the last three decades (Markotić et al., 2021). Automated radi-ology report generation aims to reduce radiologists' workload by automating image interpretation and reporting, improving efficiency and accuracy to meet growing diagnostic demands, which has attracted lots of research attention (Jing et al., 2018; Li et al., 2018; Chen et al., 2020; Wang et al., 2023a;b; Lee et al., 2023; Tu et al., 2024; Zhou et al., 2024; Wu et al., 2023; Chen et al., 2024).

In retrospect, as shown in Fig. 1 (b-1) and (b-2), there are two main paradigms in previous works. One paradigm in Fig. 1 (b-1) simplifies the process by using radiographs as the sole input modality and combining the findings and impression sections into a single long paragraph as the training target without incorporating additional clinical context. The other paradigm in Fig. 1 (b-2) leverages indication and radiographs as inputs but only generates the findings section.

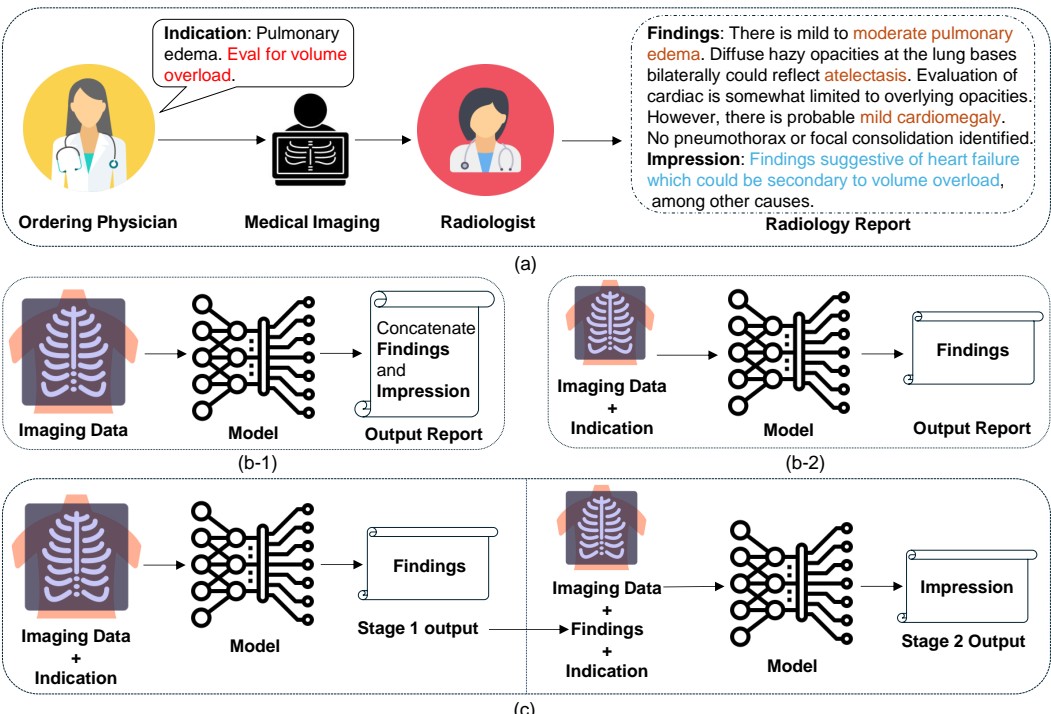

Figure 1: Comparison of real-world radiology workflow (a), two main paradigms in previous works (b-1 & b-2), and our proposed approach (c). In (a), the clinical question is highlighted in red. Findings that are closely related to the indication are highlighted in brown. The impression directly answers the clinical question (highlighted in blue). Our paradigm takes imaging data and the indication as inputs, progressively generating the findings and the impression.

Such paradigms have the following limitations: First, combining findings and the impression for training increases the model's learning difficulty due to the distinct nature of these sections. Findings typically involve detailed descriptions of specific structures and abnormalities in the image, while the impression provides a diagnosis that synthesizes these findings within the clinical context (Hartung et al., 2020). The task of generating both sections simultaneously risks the model disproportionately focusing on the more extended, more detailed findings section at the expense of overlooking the shorter but crucial impression section. As a result, the quality of impression, which require careful diagnostic reasoning, is often compromised.

Second, the content of the impression is closely related to the indication. Relying solely on imaging data hinders the model's ability to accurately capture the patterns required for generating the impression. Generating the impression is far beyond summarizing image findings but requires consideration of the patient's medical history and the reason for the examination, as illustrated in Fig. 1 (a). Without the essential context provided by the indication, the model struggles to identify which image details are the most relevant for the diagnosis. This leads to a lack of specificity and accuracy in the impression. This issue becomes particularly problematic when dealing with complex or atypical cases, where the absence of the indication makes it harder for the model to produce clinically meaningful impressions. Besides, omitting the impression section in the output, as seen in the paradigm of Fig. 1 (b-2), deviates from real-world medical practice, leaving the clinical question in the indication unanswered.

This paper tackles the above limitations by decomposing the distribution estimation processes and then conquering them progressively. Concretely, to closely align our pipeline with the workflow of radiologists', as shown in Fig. 1 (c), we break down the complex process into two successive stages: visual understanding for findings recognition followed by diagnostic reasoning. For the first stage, we only focus on transferring the information from the visual domain to the text domain, i.e., generating the findings from the given radiograph and clinical context. For the second stage, we consider what can be inferred from the radiograph that is the most relevant to the clinical question

(*i.e.*, generating the impression) aided by the output of the first stage. Finally, we directly combine the findings and the impression to yield the final reports.

Existing large-scale datasets like MIMIC-CXR are unsuitable for directly validating our proposed paradigm. Specifically, the raw data, including 227,835 free-text reports, exhibits severe inconsistencies. For instance, 57,570 reports lack the findings section, while 69,455 are missing the impression section. Even after applying MIMIC-CXR's official report parsing code, the findings section still contains misassigned contents, such as phrases that belong in the technique or comparison sections. Additionally, some reports include irrelevant information in the impression section, like "*Findings were conveyed by Dr. to at 15:33*", which is unnecessary for training report generation models. These issues create significant obstacles to verifying our approach. To address this, we propose a new benchmark derived from MIMIC-CXR, called MIMIC-1V3. It is a clean, well-structured dataset where each report is segmented into three distinct sections (*i.e.*, indication, findings, impression) and paired with one frontal radiograph. Further details are provided in Section 4.

In summary, our contributions include:

- We propose a new **R**adiologist-**L**ike **P**rogressive **G**eneration (RLPG) framework in the field of report generation, which decomposes the radiologist's workflow into visual understanding for findings recognition followed by diagnostic reasoning. Our paradigm is closer to real-world medical practice and improves the semantic alignment between input images and output reports, as demonstrated by improved clinical efficacy metrics.
- Due to no existing large-scale datasets suitable for optimizing our new paradigm, we derive a new benchmark, namely MIMIC-1V3, from the MIMIC-CXR. It serves as a standardized test bed for future report generation models.
- We conduct quantitative evaluations to assess the impact of integrating the indication as input on the quality of the generated findings and the impression. Additionally, we compare the performance of our approach against advanced LLM-based report generation models and demonstrate that it significantly outperforms them.

## 2 RELATED WORKS

The usage of ground truth reports of previous efforts in report generation diverges. Some works (Jing et al., 2018; Li et al., 2018; Chen et al., 2020; Wang et al., 2023a;b) combine findings and impression sections as the training target and take images as the sole input. In the groundbreaking work of (Jing et al., 2018), the authors explain this practice by stating: "The impression and findings sections are concatenated together as a long paragraph since impression can be viewed as a conclusion or topic sentence of the report."

Conversely, other studies focus exclusively on the findings section as the training target for given radiographs (Liu et al., 2021a; Tanida et al., 2023; Huang et al., 2023). Recent works (Zhou et al., 2021; Serra et al., 2023; Hyland et al., 2023; Bannur et al., 2024; Chaves et al., 2024) have recognized the rich clinical context in the indication section, thereby using indication and radiographs as model inputs while generating only the findings section. The exclusion of the impression section in these works is either unspecified or justified by the assertion that "the impression section summarizing the actionable insights from the study... cannot be fully gathered from the image alone."

Advancements in report generation have also seen the incorporation of large-scale training datasets and the adoption of instruction tuning techniques to build large language model (LLM)-centered multimodal multitask interactive systems. For example, RadFM (Wu et al., 2023), CheXagent(Chen et al., 2024), and MedVersa (Zhou et al., 2024) scale up the training data to the millions level, covering nearly all available public medical datasets. In their settings, the task of report generation reduces to a downstream task. CheXagent divides report generation into two tasks: (1) generating findings from the image and (2) generating impression directly from the image or through findings summarization without the image. MedVersa designs different prompts for generating findings-only, impression-only, and complete report (findings + impression). RadFM designs the prompt for report generation as "Please generate a radiology report for this scan <*image-1*>?", without specifying which sections to generate. Other notable works such as LLM-CXR (Lee et al., 2023) and R2GenGPT (Wang et al., 2023b), both obfuscate findings and impression sections, instructing the model to generate a "free-text radiology reports" or "a comprehensive and detailed diagnosis report"

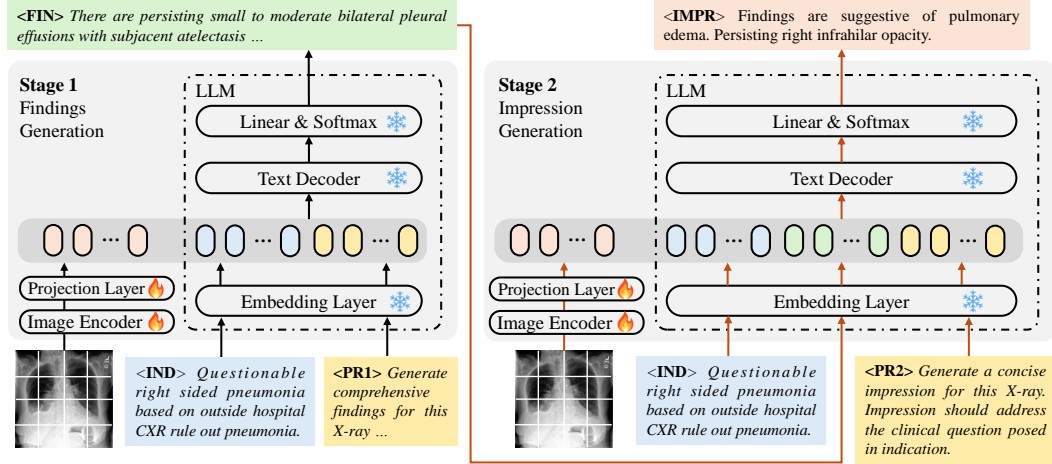

Figure 2: Overall framework. In the first stage, the inputs are image tokens and text tokens sourced from the indication (IND) and instruction prompt (PR1). The output is the findings (FIN) section. In the second stage, the inputs are image tokens combined with text tokens of the same indication, a new instruction prompt (PR2), and the findings (FIN). The output is the impression (IMPR) section.

for the given radiograph. These works mentioned above overlook the clinical context provided by the indication and fail to leverage it to guide the generation of both findings and impression.

## 3  METHOD

The overall of our proposed **R**adiologist-**L**ike **P**rogressive **G**eneration (RLPG) framework is illustrated in Fig. 2. It consists of two sequential stages, *i.e.*, findings generation and impression generation. In the first stage, the image encoder extracts the image feature into image tokens, which are concatenated with text tokens of the indication and an instruction prompt. The LLM processes these integrated tokens to yield detailed findings. In the second stage, image tokens are concatenated with text tokens of the same indication, a new instruction prompt, and the findings as the inputs of the LLM. The LLM only generates the impression in this stage. We use ground-truth findings during training and predicted findings during inference. We depict more details in the following.

### 3.1  PROBLEM REFORMULATION

In a typical radiology report generation model, the objective is to maximize the probability $p(y|x)$ between input image $x$ and output report $y$, which is a non-trivial task due to the intrinsic complicity in medical data and the large modality gap between visual inputs and textual outputs. To bridge the modality gap and tackle the intrinsic complicity in medical data, we draw inspiration from the workflow of radiologists and decompose the complex process of radiology report generation into two sequential stages and conquer them progressively. Generally, a normative radiology report $y$ consists of a findings $u$ and an impression $v$, *i.e.*, $y \leftarrow [u, v]$. In the first stage, the model takes image $x$ as input and generates only findings $u$. In the second stage, the models takes image $x$ and the generated findings $u$ as input and outputs the impression $v$. Mathematically, the probability becomes

$$p(y|x) = p(u, v|x) = p(v|x, u)p(u|x). \tag{1}$$

Moreover, to further reduce the risk of overlooking critical clinical information and enhance diagnostic accuracy, we mimic the process of radiologists interpreting the radiographs in regard to the indication. Concretely, we introduce the indication as input for both stages mentioned above. The indication directs the model to focus more intently on specific anatomical locations or abnormalities within the image $x$, ensuring a more thorough analysis. Thus, we reformulate Eq. (1) to

$$p(y|x, c) = p(u, v|x, c) = p(v|x, c, u)p(u|x, c), \tag{2}$$

where $c$ refers to the indication corresponding to paired image $x$.

## 3.2 RADIOLOGIST-LIKE PROGRESSIVE GENERATION (RLPG) FRAMEWORK

**Stage 1: Findings Generation**   As shown in Fig. 2 (left), the process begins with an image encoder $\mathcal{E}_{img}$ followed by a projection layer $\mathcal{P}^{(1)}$ encodes the radiograph $x$ into tokens that can be accepted by the LLM, capturing relevant visual features for medical diagnoses. Alongside the visual input, the indication $c$, which provides contextual clinical information, and an instruction prompt $r_1$ are used to generate textual tokens. This is done by using the embedding layer $\mathcal{E}_{txt}$ of a large language model (LLM), which prepares the text-based data for integration with the image tokens. The visual and textual tokens are concatenated to form a comprehensive input vector. This combined input is then fed through subsequent layers $\mathcal{D}_{txt}$ of the LLM, which processes the data to synthesize detailed and clinically relevant findings $u$. Mathematically, the findings generation process $\mathcal{G}^{(1)}$ can be defined as

$$u = \mathcal{G}^{(1)}(x, c, r_1) = \mathcal{D}_{txt}\left(\left[\mathcal{P}^{(1)}\left(\mathcal{E}_{img}(x)\right), \mathcal{E}_{txt}([c, r_1])\right]\right), \tag{3}$$

where $[\cdot, \cdot]$ refers to the concatenation operation. These findings encapsulate the critical observations derived from the image, which is useful for clinical decision-making.

**Stage 2: Impression Generation**   As shown in Fig. 2 (right), the same image $x$ is re-encoded using image encoder $\mathcal{E}_{img}$ with a projection layer $\mathcal{P}^{(2)}$ to ensure that any subtle features not captured during the first pass are processed. This step generates a new set of visual tokens, providing a fresh perspective on the image data. These new visual tokens are combined with the textual tokens from the same initial indication $c$, a new instructional prompt $r_2$, and the findings $u$ generated in the first stage. This comprehensive input setup ensures that the impression generation is informed by both the immediate findings and additional context that may influence the clinical interpretation. Similar to the first stage, the input then is processed by the rest of the LLM, *i.e.*, $\mathcal{D}_{txt}$. Formally, the impression generation $\mathcal{G}^{(2)}$ can be written as

$$v = \mathcal{G}^{(2)}(x, c, u, r_2) = \mathcal{D}_{txt}\left(\left[\mathcal{P}^{(2)}(\mathcal{E}_{img}(x)), \mathcal{E}_{txt}([c, u, r_2])\right]\right), \tag{4}$$

where $\mathcal{E}_{txt}$ is the embedding layer of the LLM, and $v$ is the generated impression. This impression provides the necessary contextualization and diagnostic summary to guide further medical action or evaluation.

## 3.3 TRAINING AND INFERENCE

**Training**   Based on the formulation above, our training objective can be defined as

$$\max_{\mathcal{E}_{img}, \mathcal{P}} \log p(y|x, c) = \max_{\mathcal{E}_{img}, \mathcal{P}} \log p(u, v|x, c) = \max_{\mathcal{E}_{img}, \mathcal{P}} \underbrace{\log p(v|x, c, u)}_{\text{second stage}} + \underbrace{\log p(u|x, c)}_{\text{first stage}}. \tag{5}$$

Here, we only optimize the image encoder $\mathcal{E}_{img}$ and projection layer $\mathcal{P}$ while keeping the LLM parameters fixed. This way refines visual feature extraction for our specific needs without disturbing the established linguistic capabilities of the LLM, ensuring stable text generation. Notably, we also omit the instruction prompts $r_1$ and $r_2$ since they are consistent across all samples during both the training and inference phases.

In the first stage, we initialize the image encoder $\mathcal{E}_{img}$ with ImageNet pre-trained weights and randomly initialize the projection layer $\mathcal{P}$. In general, sequence generation models are often trained using the autoregressive Teacher Forcing technique, which maximizes the likelihood of the often token $w_t$ given all previous often tokens $w_{i<t}$. In our setting, we optimize the model by

$$\mathcal{L}_1 = -\log p(u|x, c) = \sum_{t=1}^{T} -\log p(w_t|w_{i<t}, x, c), \tag{6}$$

where $w_t$ is the $t$-th token in the findings $u$ and $T$ is the total number of tokens in $u$.

We take the well-trained image encoder in the first stage as an initialization for the second-stage image encoder with a new randomly initialized projection layer. Similar to the optimization method in stage one, the loss function in the second stage can be written as

$$\mathcal{L}_2 = -\log p(v|x, c, u) = \sum_{l=1}^{L} -\log p(w_l|w_{j<l}, x, c, u), \tag{7}$$

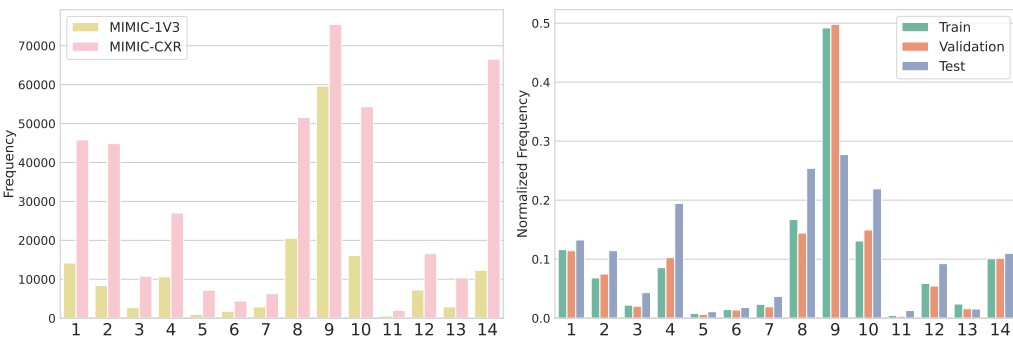

(a) Positively mentioned labels in MIMIC-CXR and MIMIC-1V3.     (b) Normalized Label distribution across splits in MIMIC-1V3.

Figure 3: Each index represents a specific label defined in CheXpert. 1: Atelectasis, 2: Cardiomegaly, 3: Consolidation, 4: Edema, 5: Enlarged Cardiomediastinum, 6: Fracture, 7: Lung Lesion, 8: Lung Opacity, 9: No Finding, 10: Pleural Effusion, 11: Pleural Other, 12: Pneumonia, 13: Pneumothorax, 14: Support Devices.

where $w_l$ is the $l$-th token in the impression $v$ and $L$ is the total number of tokens in $v$. Note that we use ground-truth findings during training and the predicted findings in inference.

**Inference** In the first stage, with the pre-defined instruction prompt $r_1$, the whole model $\mathcal{G}^{(1)}$ takes a radiograph $x$ and the indication $c$ as inputs to generate findings $\tilde{u}$. In the second stage, with another prompt $r_2$, both the generated findings $\tilde{u}$ and the same inputs $x$ and $c$ are fed into model $\mathcal{G}^{(2)}$ to produce an impression $\tilde{v}$. The final diagnostic report $\tilde{y}$ is then composed of these two outputs, *i.e.*, $\tilde{y} \leftarrow [\tilde{u}, \tilde{v}]$. Formally, the whole inference process can be defined as

$$\tilde{y} \leftarrow [\tilde{u}, \tilde{v}], \text{ where } \tilde{u} = \mathcal{G}^{(1)}(x, c, r_1) \text{ and } \tilde{v} = \mathcal{G}^{(2)}(x, c, \tilde{u}, r_2). \tag{8}$$

This structured, sequential approach allows each model to specialize in distinct aspects of report generation, enhancing the overall accuracy and relevance of the generated reports.

## 4 MIMIC-1V3 DATASET

### 4.1 MIMIC-1V3 DATASET CONSTRUCTION

Raw reports from MIMIC-CXR are unstructured with varying numbers of sections. For example, 12.5% of all reports do not have the findings section. We aim to provide a clean and structured dataset with easy access to different report sections. The construction of MIMIC-1V3 mainly comprises three steps:

**View Selection** The MIMIC-CXR dataset contains 14 unique X-ray image views. The frontal view, comprising both Anterior-Posterior (AP) and Posterior-Anterior (PA) views, makes up 64.5% of all images. The number of images paired with each report varies, with 45.4% of reports associated with just one image. We exclusively use one frontal view image for consistency across samples for each paired report.

**Expanding Abbreviations & Acronyms** Abbreviations and acronyms frequently appear in the indication section. We obtain standardized medical abbreviations from radiopaedia [1] and automatically expand them.

**Report Parsing & Cleaning** We use MIMIC-CXR's official code base to parse free text reports, extracting the indication, findings, and impression sections while excluding the incorrectly categorized *comparison* and *technique* sections. For example, phrases like "*Two frontal chest radiographs were obtained with patient positioned upright*" are mistakenly placed under findings instead of technique. To provide a cleaner version of the report, we manually review approximately 20,000 reports to identify patterned phrases and then automatically remove such misassigned phrases from other reports. Additionally, we discard reports lacking all three sections, for example, 59,628 reports contain only

---

[1]https://radiopaedia.org/

```
Human: <ImageHere></img>.
Given patient's indication: {###}.
Generate a detailed and
comprehensive findings section for this
X-ray examination. Findings are the
factual observation of the X-ray image.
Assistant:
```

```
Human: <ImageHere></img>.
Given patient's indication: {###} and
findings: {###}. Generate a concise
impression section for this X-ray
examination. Impression should address
the clinical question posed in indication.
Assistant:
```

(a)                                          (b)

Figure 4: Prompts for (a) Stage 1 and (b) Stage 2.

indication and impression, leaving it unclear whether the findings section is absent or misassigned during data collection.

### 4.2 MIMIC-1V3 DATASET ANALYSIS

**Data Split and Distribution**   MIMIC-CXR includes an official CheXpert (Irvin et al., 2019) label file for all reports, in which the value for the positively mentioned labels is set to 1. Retaining adequate positively mentioned labels in MIMIC-1V3 is crucial for providing strong and unambiguous supervision for learning the image-text alignment.

Specifically, our MIMIC-1V3 results in 119,395 training, 936 validation, and 1,546 test samples. We strictly follow the official data partition of MIMIC-CXR to ensure that our training, validation, and test samples are sourced exclusively from the original sets. Fig. 3a shows the overall distribution of positively mentioned labels in MIMIC-1V3 and MIMIC-CXR.

Compared to the original dataset, six labels in MIMIC-1V3 maintain at least 40% positive mentions. On the other hand, *Support Devices* maintains only 19.1% of original positive mentions and 17% for *Enlarged Cardiomediastinum*. We consider the impact of lacking *Support Device* to be limited, as the presence of medical devices is not systematically reported in clinical practice by radiologists (Bustos et al., 2020). An insufficient amount of *Enlarged Cardiomediastinum* remains an unresolved challenge. The label distribution in the training, validation, and test sets is shown in Fig. 3b. The training and validation sets have similar label distributions, while there is a noticeable discrepancy between the label distributions in the validation and test sets.

**Lengths of Indication, Findings, and Impression**
Table 1 shows the word count for three sections across all dataset splits. The length of the indication section remains consistent across splits. Both training and validation sets follow a similar distribution. In contrast, the average length of findings and impression sections on the test set are 22% and 25% longer than those on the training and validation sets, respectively.

Table 1: Word counts of the indication, findings and impression in all splits with standard deviation on our MIMIC-1V3.

| Split | Indication | Findings | Impression |
|---|---|---|---|
| Train | $9.89 \pm 6.67$ | $45.51 \pm 21.49$ | $12.96 \pm 6.17$ |
| Validation | $10.23 \pm 6.69$ | $45.85 \pm 22.50$ | $12.98 \pm 6.86$ |
| Test | $9.94 \pm 6.74$ | $58.20 \pm 23.76$ | $16.82 \pm 13.19$ |

## 5 EXPERIMENTS

### 5.1 IMPLEMENTATION DETAILS

During training, for both stages, we employ Swin Transformer (Liu et al., 2021b)-base as the image encoder, one linear layer as the projection layer, and Llama2-7B-chat (Touvron et al., 2023) as our LLM. The image encoders and projection layers are trainable, while the LLM remains frozen and shared across both stages. Fig. 4 demonstrates the different prompts we used for each stage. For stage 1, we instruct the LLM to generate findings conditioned on image and indication and explicitly define the findings as the factual observation of the X-ray image. For stage 2, we include the findings as part of the prompt. The LLM is instructed to generate the impression based on the image, indication, and findings, focusing on addressing the clinical question posed in the indication.

Table 2: Model performance of baseline one-stage training and our progressive paradigm. Here, we abbreviate image, indication, findings, and impression as IMG, IND, FIN, and IMPR. Symbols indicate the following: ✓ = using images or ground truth findings as input, ✓ = using inference findings as input, and ✓ = sections that are model outputs.

| Dataset | Split | Input | | | Output | | Findings | | | | | | | Impression | | | | | | |
|---|---|---|---|---|---|---|---|---|---|---|---|---|---|---|---|---|---|---|---|---|
| | | IMG | IND | FIN | FIN | IMPR | NLG | | Clinical Efficacy | | | | | NLG | | Clinical Efficacy | | | | |
| | | | | | | | B-4 | CIDEr | Bert-S | F1-all | RE-EM | RE-NLI | Rad-C | B-4 | CIDEr | Bert-S | F1-all | RE-EM | RE-NLI | Rad-C |
| MIMIC-1V3 | val | ✓ | | | ✓ | ✓ | 0.1169 | 0.3084 | 0.5744 | 0.4041 | 0.3884 | 0.3238 | 0.2285 | 0.0429 | 0.2726 | 0.3518 | 0.5025 | 0.2310 | 0.2803 | 0.1187 |
| | | ✓ | | | ✓ | | 0.1588 | 0.4293 | 0.5766 | 0.4617 | 0.4095 | 0.3619 | 0.2425 | - | - | - | - | - | - | - |
| | | ✓ | | ✓ | | ✓ | - | - | - | - | - | - | - | 0.0824 | 1.1259 | 0.4787 | 0.5057 | 0.3017 | 0.3403 | 0.2011 |
| | | ✓ | | ✓ | | ✓ | - | - | - | - | - | - | - | 0.2430 | 2.9432 | 0.6342 | 0.7125 | 0.5439 | 0.5246 | 0.4139 |
| | test | ✓ | | | ✓ | ✓ | 0.1044 | 0.1530 | 0.5595 | 0.4476 | 0.3631 | 0.2363 | 0.1981 | 0.0532 | 0.4530 | 0.3277 | 0.3875 | 0.1539 | 0.1657 | 0.0869 |
| | | ✓ | | | ✓ | | 0.1139 | 0.1663 | 0.5437 | 0.4225 | 0.3320 | 0.2360 | 0.1873 | - | - | - | - | - | - | - |
| | | ✓ | | ✓ | | ✓ | - | - | - | - | - | - | - | 0.0582 | 0.6060 | 0.4411 | 0.4301 | 0.2374 | 0.1686 | 0.1481 |
| | | ✓ | | ✓ | | ✓ | - | - | - | - | - | - | - | 0.1948 | 1.5391 | 0.5715 | 0.6475 | 0.4504 | 0.2853 | 0.3081 |

We use AdamW (Loshchilov, 2017) as the optimizer and cosine scheduler with a learning rate of 3e-5 and a weight decay of 0.01. The majority of our experiments are conducted on two L40S GPUs. During inference, the model first generates *findings* based on image and indication. Then we use generated *findings*, image, and indication to generate the impression.

## 5.2 EVALUATION METRICS

For Natural Language Generation (NLG) metrics, we use BLEU-4 (B-4) (Papineni et al., 2002) and CIDEr (Vedantam et al., 2015). For Clinical Evaluation (CE) metrics, we employ Bert-Score (Bert-S) (Zhang et al., 2019), Chexbert-all Micro F1 (F1-all) (Smit et al., 2020), and RadGraph-related metrics (Jain et al., 2021): RadEntity-ExactMatch (RE-ME), RadEntity-NLI (RE-NLI) (Miura et al., 2020), and RadGraph-Complete (Rad-C).

## 5.3 ANALYSIS OF EVALUATION RESULTS

We thoroughly analyze evaluation results on both validation and test sets. Results in Table 2 and Table 3 are obtained with the same model architecture. The differences among the settings are the input data, output sections, and training stages (*i.e.*, one stage or our progressive stages).

**Advantages of Our Progressive Training** In Table 2, we provide the NLG and CE metrics of baseline (*i.e.*, one-stage training) and our proposed progressive paradigm. In the baseline setting, the model receives only the image as input and simultaneously generates both the findings and impression sections. In contrast, our progressive paradigm operates in two stages: first, the model takes the image as input and generates only the findings section; second, it takes the generated findings and the original image to produce the impression section. We evaluate each section of the output separately.

On the validation set, the performance of both the findings and impression sections is pronounced. In the baseline, the findings section achieves a CIDEr score of 0.3084 and an F1-all score of 0.4041. However, the performance for the impression section is lower, with a CIDEr score of 0.2726 and a Bert-Score of 0.3518, indicating that the baseline model exhibits greater difficulty in generating the impression section. In contrast, with our proposed progressive paradigm, the model's performance constantly improves, especially in generating the impression section. On the val split, the CIDEr score of the findings increases from 0.3084 to 0.4293 and the F1-all score increases from 0.4041 to 0.4617. More notably, for the impression section, when using predicted findings as input, the CIDEr score increases from 0.2726 to 1.1259, and the Bert-Score increases from 0.3518 to 0.4787. Elevated evaluation scores prove our progressive generation strategy can enhance clinical efficacy and text generation quality.

Despite the significant label frequency and section length shift between the training and test sets, as illustrated in Fig. 3b and Table 1, which causes a noticeable performance discrepancy between validation and test sets, we still observe the advantages of our progressive paradigm on the test set. Specifically, with the progressive paradigm, the findings generation CIDEr score increases from 0.1530 in the baseline to 0.1663. For the impression section, the CIDEr score improves from 0.4530 to 0.6060, and the F1-all score rises from 0.3875 to 0.4301. These results highlight that, despite

Table 3: Evaluation results when using the indication as part of the input. For the first stage, the model takes the image and the indication as input and outputs the findings section. In the second stage, the model takes an image, the indication, and the predicted findings as inputs and outputs the impression section.

| Dataset | Split | Input | | | Output | | Findings | | | | | | | Impression | | | | | | |
|---|---|---|---|---|---|---|---|---|---|---|---|---|---|---|---|---|---|---|---|---|
| | | IMG | IND | FIN | FIN | IMPR | NLG | | Clinical Efficacy | | | | | NLG | | Clinical Efficacy | | | | |
| | | | | | | | B-4 | CIDEr | Bert-S | F1-all | RE-EM | RE-NLI | Rad-C | B-4 | CIDEr | Bert-S | F1-all | RE-EM | RE-NLI | Rad-C |
| MIMIC-1V3 | val | ✓ | | | ✓ | | 0.1588 | 0.4293 | 0.5766 | 0.4617 | 0.4095 | 0.3619 | 0.2425 | - | - | - | - | - | - | - |
| | | ✓ | ✓ | | ✓ | | 0.1825 | 0.6376 | 0.5906 | 0.4723 | 0.4192 | 0.3569 | 0.2554 | - | - | - | - | - | - | - |
| | | ✓ | | ✓ | | ✓ | - | - | - | - | - | - | - | 0.0824 | 1.1259 | 0.4787 | 0.5057 | 0.3017 | 0.3403 | 0.2011 |
| | | ✓ | ✓ | ✓ | | ✓ | - | - | - | - | - | - | - | 0.1075 | 1.6675 | 0.5239 | 0.5475 | 0.3647 | 0.4159 | 0.2499 |
| | | ✓ | | ✓ | | ✓ | - | - | - | - | - | - | - | 0.2430 | 2.9432 | 0.6342 | 0.7125 | 0.5439 | 0.5246 | 0.4139 |
| | | ✓ | ✓ | ✓ | | ✓ | - | - | - | - | - | - | - | 0.2705 | 3.1813 | 0.6485 | 0.7133 | 0.5569 | 0.5830 | 0.4252 |
| | test | ✓ | | | ✓ | | 0.1139 | 0.1663 | 0.5437 | 0.4225 | 0.3320 | 0.2360 | 0.1873 | - | - | - | - | - | - | - |
| | | ✓ | ✓ | | ✓ | | 0.1155 | 0.2856 | 0.5630 | 0.4516 | 0.3685 | 0.2430 | 0.2026 | - | - | - | - | - | - | - |
| | | ✓ | | ✓ | | ✓ | - | - | - | - | - | - | - | 0.0582 | 0.6060 | 0.4411 | 0.4301 | 0.2374 | 0.1686 | 0.1481 |
| | | ✓ | ✓ | ✓ | | ✓ | - | - | - | - | - | - | - | 0.0648 | 0.7547 | 0.4581 | 0.4308 | 0.2773 | 0.2206 | 0.1727 |
| | | ✓ | | ✓ | | ✓ | - | - | - | - | - | - | - | 0.1948 | 1.5391 | 0.5715 | 0.6475 | 0.4504 | 0.2853 | 0.3081 |
| | | ✓ | ✓ | ✓ | | ✓ | - | - | - | - | - | - | - | 0.2074 | 1.6740 | 0.5897 | 0.6614 | 0.4892 | 0.3513 | 0.3306 |

the distributional shifts between splits, the progressive paradigm demonstrates stronger generalization ability, particularly in generating clinically meaningful impressions. In addition, on validation and test sets, replacing the predicted findings with ground-truth findings for impression generation results in a considerable leap across all metrics. This denotes that the quality of the findings dramatically impacts the impression quality. Overall, even though the distributional shift affects the test set performance, the progressive model consistently shows its superiority in generating higher-quality findings and impressions, further proving the effectiveness of our progressive generation strategy for radiology report generation.

**Benefits of Using Indication as Input**  In this part, we explore the benefits of including the indication as input in our progressive framework. To this end, we compare the performance of our framework with and without inputting the indication. As shown in Table 3, on the validation set, after incorporating the indication as inputs, the BLEU-4 and CIDEr scores of the findings increase from 0.1588 to 0.1825 and from 0.4293 to 0.6376, respectively. All CE scores have improved except the RadEntity-NLI (with comparable performance).

The benefit of using the indication as input is more evident on the generation of impression than on findings. The improvement can be observed across all evaluation metrics, such as CIDEr score increases from 1.1259 to 1.6675 and RadEntity-NLI score increase from 0.3403 to 0.4159. After replacing the predicted findings with ground-truth findings for impression generation, the indication can still further enhance the quality of the impression, improving BLEU-4 score from 0.2430 to 0.2705, CIDEr score from 2.9432 to 3.1813, and RadEntity-NLI score from 0.5246 to 0.5830.

On the test set, despite performance drops compared to the validation set due to different label distributions as shown in Fig. 3, incorporating the indication as input leads to notable improvements in findings generation. Specifically, the CIDEr score increases from 0.1663 to 0.2856, the RadEntity-ExactMatch score rises from 0.3320 to 0.3685, and the RadGraph-Complete score improves to 0.2026. For impression generation, incorporating the indication results in an increase in the BLEU-4 score from 0.0582 to 0.0648, the CIDEr score from 0.6060 to 0.7547, the RadEntity-NLI score from 0.1686 to 0.2206, and the RadGraph-Complete score rises to 0.1727.

These improvements are due to the indication serving as a guideline for the generation of findings, providing context and a soft boundary for what should be considered in the findings that are closely related to the patient's medical history and the clinical question. The indication and impression are similar to a question-answer pair where the indication raises the question, and the impression addresses it. Using the indication as the input could provide a solid guiding signal.

**Comparison with Other LLM-based Report Generation Models**  We select three representative LLM-based models (*i.e.*, R2GenGPT (Wang et al., 2023b), RadFM (Wu et al., 2023), and CheXagent (Chen et al., 2024)) to perform inference directly on the MIMIC-1V3 test set. For RadFM, we use the prompt: "*Can you provide a caption that consists of both findings and impression for this medical image?*" For CheXagent, we follow the prompts specified in their original paper: (1) Given the <*image*>, generate its <*findings*>; (2) Given the <*image*>, generate its <*impression*>; and (3) Given the <*findings*>, generate its <*impression*>.

Table 4: Comparison with other LLM-based RRG methods.

| Dataset | Method | Input | | | Output | | Findings | | | | | | | Impression | | | | | | |
| | | IMG | IND | FIN | FIN | IMPR | NLG | | Clinical Efficacy | | | | | NLG | | Clinical Efficacy | | | | |
| | | | | | | | B-4 | CIDEr | Bert-S | F1-all | RE-EM | RE-NLI | Rad-C | B-4 | CIDEr | Bert-S | F1-all | RE-EM | RE-NLI | Rad-C |
| MIMIC-1V3 | R2GenGPT | ✓ | | | ✓ | ✓ | 0.1044 | 0.1530 | 0.5595 | 0.4476 | 0.3631 | 0.2363 | 0.1981 | 0.0532 | 0.4530 | 0.3277 | 0.3875 | 0.1539 | 0.1657 | 0.0869 |
| | RadFM | ✓ | | | ✓ | ✓ | 0.0003 | 0.0110 | 0.1413 | 0.1582 | 0.1005 | 0.0589 | 0.0434 | 0.0041 | 0.1783 | 0.1526 | 0.2423 | 0.0643 | 0.0512 | 0.0397 |
| | CheXagent | ✓ | | | ✓ | | 0.0540 | 0.0732 | 0.5152 | 0.2588 | 0.3484 | 0.2639 | 0.1740 | - | - | - | - | - | - | - |
| | CheXagent | ✓ | | | | ✓ | - | - | - | - | - | - | - | 0.0305 | 0.6147 | 0.3807 | 0.3749 | 0.2169 | 0.1753 | 0.1323 |
| | CheXagent | | | ✓ | | ✓ | - | - | - | - | - | - | - | 0.0229 | 0.4732 | 0.3881 | 0.3075 | 0.1496 | 0.1624 | 0.0928 |
| | Ours | ✓ | ✓ | | ✓ | | 0.1155 | 0.2856 | 0.5630 | 0.4516 | 0.3685 | 0.2430 | 0.2026 | - | - | - | - | - | - | - |
| | Ours | ✓ | ✓ | ✓ | | ✓ | - | - | - | - | - | - | - | 0.0648 | 0.7547 | 0.4581 | 0.4308 | 0.2773 | 0.2206 | 0.1727 |

As shown in Table 4, our framework outperforms the baseline LLM-based models by a large margin. RadFM shows limited effectiveness in generating clinically meaningful findings, achieving a BLEU-4 score of 0.0003 and a RadGraph-Complete score of 0.0434. However, it demonstrates slightly better performance in impression generation than in findings generation, likely due to the brevity of the impression section.

Although CheXagent achieves improved evaluation results in the findings section, it remains inferior to our framework, obtaining BLEU-4 scores of 0.0540 versus 0.1155, CIDEr scores of 0.0732 versus 0.2856, and F1-all scores of 0.2588 versus 0.4516. Conversely, for CheXagent, generating the impression directly from the image yields better results than generating the impression by summarizing its own generated findings. This observation suggests that the findings produced by CheXagent are less informative than the original image data. Consequently, incorporating the image as input for impression generation proves beneficial, particularly when the quality of the generated findings is sub-optimal.

While RadFM and CheXagent can perform multiple tasks and handle modalities beyond X-rays, R2GenGPT is exclusively dedicated to chest X-ray report generation. Consequently, it is unsurprising that R2GenGPT's performance is comparable to our proposed approach. It is important to highlight that RadFM employs a domain-specific LLM comprising 13B parameters, whereas CheXagent fine-tunes a Mistral-7B model on their medical corpus. In contrast, our model utilizes a standard Llama2-7B-chat. By incorporating the indication as input and adopting a progressive generation strategy, our approach effectively compensates for the lack of domain-specific knowledge within the base LLM.

In summary, we demonstrate the quantitative impact of integrating indication as inputs on the quality of generated findings and impression. We can conclude that: 1) our radiologist-like progressive generation paradigm *i.e.*, successively generating findings and the impression, is more effective than generating both sections at once, and 2) incorporating indication as part of input can benefit the generation of findings and impression on both NLG metrics and CE metrics.

## 6 CONCLUSION AND FUTURE WORKS

In this work, we propose a novel radiologist-like progressive generation (RLPG) framework for automated radiology report generation, consisting of two successive stages: visual understanding for findings recognition followed by diagnostic reasoning. In each stage, we incorporate the indication as input, closely mimicking real-world radiology workflows and resulting in more clinically accurate reports than other LLM-based report generation models. Besides, we introduce a new benchmark, MIMIC-1v3, derived from MIMIC-CXR. In MIMIC-1v3, each report is segmented into three sections—indication, findings, and impression—and paired with the radiograph. Compared to MIMIC-CXR, our dataset is cleaner and highly structured, ensuring consistency across all samples.

Future research could expand the framework and dataset to address different radiological sub-specialties, enhance natural language understanding for better clinical language interpretation, and incorporate temporal analysis to track patient condition changes over time. Testing the framework's adaptability across diverse datasets, especially from different anatomical regions or healthcare systems, will ensure its generalizability and robustness. Moreover, enriching the MIMIC-1v3 dataset with more detailed annotations, such as disease severity, could increase the utility and clinical relevance of the automated reports.

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

# A APPENDIX

## A.1 SCOPE OF DIFFERENT SECTIONS IN A REPORT

The problem formulation of report generation has been deemed similar to image captioning, i.e., describing salient objects in coherent sentences. However, captioning a natural image falls well within the scope of common sense. In contrast, interpreting a medical image takes years of professional training, requiring radiologists to draw upon their full depth of knowledge and experience to deliver meaningful patient care. As the most important product of medical imaging, the radiology report represents the sum of a radiologist's highest level of synthesis and insight into a patient's condition. A clinically acceptable report communicates the diagnosis or differential diagnosis, clinical implications of radiologic findings, and recommendations for directing patient management (Hartung et al., 2020). The complexity of dictating a radiology report, which transcends simply describing factual observations in the radiograph, warrants a more profound and clinical inspection of the dataset.

A typical radiology report is structured into several distinct sections, each with a defined purpose (Collard et al., 2014; Hartung et al., 2020; Wallis & McCoubrie, 2011). As regional and personal stylistic preferences abound, the naming convention of sections may vary. For example, in MIMIC-CXR, the indication is sometimes named clinical history. We unify the naming convention based on the most frequent terms for a more succinct and clear structure. We list the seven most prevalent sections and clearly define the scope of each section based on established medical literature as below:

1. **Wet Read**: A preliminary interpretation of the radiograph without in-depth analysis, often used for urgent cases at point-of-care.
2. **Comparison**: This section notes the availability of previous studies, enabling radiologists to monitor patient's progress. At times, reaching the conclusion that whether findings are benign or malignant may require a thorough evaluation of comparison studies.
3. **Technique**: This section documents information about the imaging modality used, the specific imaging parameters, specific projection views used in imaging such as "supine AP" and any additional details relevant to the acquisition of the images such as subpar imaging quality due to patient position.
4. **Indication**: It presents the clinical question prompting the examination (reason for this exam) and offers a brief overview of the patient's medical history.
5. **Findings**: The radiologist records factual observations from the radiograph. It comprises short informative phrases describing the pertinent positive and negative observations about a study. Findings emphasize facts and should avoid interpretation or synthesis intended for the impression.
6. **Impression**: It provides a diagnosis or differential diagnosis (a definitive diagnosis may be out of reach because of inherent limitations of the X-ray modality) when possible, followed by the key findings relevant to understanding the extent of the disease. It is the sum of all the efforts in synthesizing the meaning of findings and answering the clinical question raised in the *indication*.
7. **Recommendation(s)**: This includes the radiologist's opinion on directing patient care.

The Wet Read section lacks reporting maturity as a preliminary report, which is excluded from our study. The header portion (comparison and techniques) as auxiliary information is considered beyond the scope of our work. Their function is self-evident in that the comparison is related to longitude information, and the techniques can be used for view classification. The Recommendation(s) section is also excluded from our study as it is not closely related to interpreting images.

| Study_ID: 52093225 | Study_ID: 54495391 |
|---|---|
| **indication**: Hypoxia, chest pain, dyspnea, question infiltrate. **findings**: There is patchy opacity with air bronchograms at both lung bases, consistent with a pneumonic infiltrate. The differential diagnosis could include aspiration, but this is considered less likely. There do appear to be background increased interstitial markings which could be related to either acute or chronic lung disease. Cardiomediastinal silhouette is slightly prominent, but likely accentuated by **low lung volumes**. The **right hemidiaphragm** is **elevated**. Mild prominence of the **azygos vein** is likely also accentuated by low lung volumes. **impression**: Findings concerning for bilateral pneumonic infiltrates. Also diffusely increased interstitial markings, which may indicate a background acute or chronic process. | **indication**: M with complaints of left lower chest pain with shortness of breath and cough.? pneumonia **findings**: Left chest wall single lead pacing device is again seen. **Low lung volumes** are noted. Increased interstitial markings are noted in the lungs with a basilar predominance which are compatible with a chronic interstitial abnormality. There is no superimposed acute consolidation or effusion. The cardiomediastinal silhouette is stable. No acute osseous abnormalities. Hypertrophic changes are seen the spine. **impression**: Findings compatible with patient's underlying fibrosis without definite superimposed acute cardiopulmonary process. |
| (a) | (b) |

Figure 5: Two representative examples from MIMIC-1V3. The highlighted parts demonstrate the interconnections among the *indication*, *findings*, and *impression* sections.

Here, we focus on discussing the relationships and connections among the indication, findings, and impression.

From a clinical perspective, a radiology report is a bridge between the ordering physician, the radiologist, and the referring clinician, communicating critical patient information. In the indication, the ordering physician raises a clinical question based on the patient's current condition and medical history. This question guides the radiologist's attention, enabling the radiology to focus on the most pertinent anatomic locations. The findings section is for factual observations about the image and reflects the radiologist's thought process. The impression section is more interpretive, drawing on the radiologist's expertise to infer conclusions from the findings. For instance, lung opacity refers to an objective observation; consolidation is commonly used to describe an opacity that may resemble pneumonia, and pneumonia is a clinical inference (Wu et al., 2020). The impression should answer the clinical question, providing a context for the referring clinician to understand the implications of radiologic findings.

## A.2 CASE STUDIES OF MIMIC-1V3

In Fig. 5, two reports from MIMIC-1V3 demonstrate how real-world data reflects the connections among the indication, findings, and impression. The pattern is clear: the indication presents a clinical question (highlighted in red), the findings section contains observations closely related to the clinical question (highlighted in brown), and the impression directly addresses the clinical question followed by primary findings (highlighted in blue).

In report (a), the indication states the patient's condition as **Hypoxia, chest pain, dyspnea**, meaning the patient has difficulty breathing. These symptoms prompt the radiologist to assess the respiratory and cardiovascular structures for abnormalities that could explain the patient's distress. **Question Infiltrate**, this specific clinical query directs the radiologist to scrutinize the lung parenchyma for signs of **infiltrates**, such as **patchy opacity**. **Elevated Right Hemidiaphragm** and **Prominent Azygos Vein** are the secondary signs that may relate to the primary symptoms of **dyspnea** and **hypoxia**. The impression section confirms the presence of bilateral **pneumonic infiltrates**, directly answering the clinical question posed in the indication. It also highlights other notable observations from the imaging study, ensuring that other potential issues are not overlooked and assisting the referring clinician in understanding the full scope of the patient's pulmonary condition.

In report (b), the original indication is presented as "M with c/o left lower CP with SOB and cough. ? PNA." The extensive use of abbreviations and acronyms renders the indication ambiguous and unclear. Expanding these abbreviations and acronyms to their complete forms provides a more precise context, thereby reducing model hallucinations during report generation. This issue has been addressed in the construction of MIMIC-1V3, where abbreviations have been standardized and expanded.

## A.3 VISUALIZATION OF GENERATED REPORTS

To further demonstrate the superiority of our framework over other LLM-based models, we showcase how our framework effectively handles the clinical query in the indication and generates a

| MIMIC-1V3 | Ground Truth | R2GenGPT | RadFM | CheXagent | RLPG (Ours) |
|---|---|---|---|---|---|
| **Study_ID: 52526911**

**Indication**: -year-old female with altered mental status and history of seizures. Evaluate for aspiration or pneumonia. | **Findings**: Lung volumes are very low and marked elevation of the left greater than right hemidiaphragm is similar to prior. Bibasilar atelectasis is unchanged. Cardiomediastinal contours are stable. Pulmonary vascular markings appear normal. No focal consolidation or large pleural effusion.

**Impression**: Low lung volumes without radiographic evidence of aspiration or focal consolidation. | **Findings**: There are low lung volumes with bibasilar atelectasis. The cardiomediastinal silhouette is not well assessed due to low lung volumes. There is no large pleural effusion or pneumothorax. Bony structures appear grossly unremarkable.

**Impression**: There are low lung volumes with bibasilar atelectasis. | **Findings**:Right pleural effusion. Large mass projects over the right upper chest.

**Impression**: Right pleural effusion. Large mass in the right upper chest. | **Findings**: The heart size is normal. The mediastinal and hilar contours are unremarkable. The pulmonary vasculature is normal. The lungs are clear. No pleural effusion or pneumothorax is seen. There are no acute osseous abnormalities.

**Impression**: No acute cardiopulmonary process. | **Findings**: Lung volumes are low. The cardiac mediastinal and hilar contours appear unchanged. There is crowding of the bronchovascular markings without definite focal consolidation. no pleural effusion or pneumothorax is seen. There is bibasilar atelectasis.

**Impression**: Low lung volumes with bibasilar atelectasis. No definite evidence for aspiration or pneumonia. |

Figure 6: Visualization of generated reports

higher-quality report. The most relevant contents among the indication, findings, and impression are highlighted in red, while secondary findings that are worth mentioning are underlined.

As depicted in Fig. 6, the ground truth indication states, "Evaluate for aspiration or pneumonia," which requires the model to make an accurate diagnosis. The performance of RadFM is inferior; it misdiagnoses the patient with right pleural effusion and repeats most findings in the impression. CheXagent is the only model that mentions pulmonary vasculature in the report, but it fails to identify the key finding of "low lung volumes." Since CheXagent treats the impression as the summarization of the findings and does not take the indication as the input, its generated impression, "No acute cardiopulmonary process," lacks the ability to address the clinical query and only reflects the fact that all its predicted findings are negative.

R2GenGPT and our framework can identify the key finding, "low volumes." However, without the indication as input, R2GenGPT tends to repeat the positive findings in the impression, which falls short of effectively addressing the clinical query.

In contrast, our framework correctly predicts "without definite focal consolidation" in the findings and generates an impression that addresses the clinical query with a sound diagnosis, " No definite evidence for aspiration or pneumonia," which slightly deviates from the ground truth but matches the semantic meanings.

Regrettably, all models fail to identify conditions related to the hemidiaphragm. Enhancing the model's image understanding capabilities could help mitigate such errors.

