# OpenReview forum: "Radiologist-like Progressive Radiology Report Generation and Benchmarking"
_ICLR.cc/2025/Conference — Submitted to ICLR 2025_

### Official Review · Reviewer_t2eU · 2024-10-31

**Soundness:** 3
**Presentation:** 4
**Contribution:** 2
**Rating:** 5
**Confidence:** 5

**Summary:**

This paper emulates the workflow of radiologists to structure the report generation task in a sequence of indication + image → findings → impression, proposing a multi-stage generation framework. Additionally, it curates reports of the MIMIC-CXR, the largest chest X-ray dataset, into three sections: indication, findings, and impression.

**Strengths:**

1. Medical significance: This paper divides lengthy reports into three main sections to emulate radiologists' workflow, providing insights into the reasoning of large language models and identifying the stages that may contribute to erroneous reports.
2. Clarity: The statistics of the created MIMIC-1V3 are provided. The figures and tables are informative and easy to understand. The proposed 2-stage report generation algorithm is easy to understand.

**Weaknesses:**

1. Originality and technical novelty: The proposed Radiologist-Like Progressive Generation (RLRG) is an MLLM-based 2-stage framework that feeds the output of the first stage as input of the second stage. Therefore, the overall design is not technically novel. Further, the neural network architecture is similar to existing radiology report generation methods such as R2Gen; the proposed dataset is modified from the existing publicly accessible dataset.
2. Clarity: The authors concentrated on describing the 2-stage report generation algorithm, while the details of curating the MIMIC-1V3 dataset are not clearly stated. As the contribution of this paper heavily relies on the dataset creation, it is better to clearly address questions such as “How to handle reports without impression/finding/induction?”, “what algorithm was used to separate the report?” and “How to evaluate the correctness of the split?”.

**Questions:**

1. Please further highlight the technical contribution of the proposed RLRG framework, such as differences compared to existing multi-stage generation or MLLMs.
2. From an application of MLLM’s perspective, there is not much difference between indication + image -> finding & impression and indication + image -> finding -> impression, as both findings and impressions are important and should be generated as medical reports. Please further explain the need to split the findings and impressions.

---

> ### Author Response · Authors · 2024-11-23
> **Response to Reviewer t2eU (1/2)**
>
> Dear Reviewer t2eU,
>
> Thank you for your detailed and constructive feedback on our work. We appreciate your insights into both the strengths and weaknesses of our approach, and we address each of your points below:
>
> **Weaknesses**
>
> **Q1**: Originality and technical novelty: The proposed Radiologist-Like Progressive Generation (RLRG) is an MLLM-based 2-stage framework that feeds the output of the first stage as input of the second stage. Therefore, the overall design is not technically novel. Further, the neural network architecture is similar to existing radiology report generation methods such as R2Gen; the proposed dataset is modified from the existing publicly accessible dataset.
>
> **A**: The main contribution of our work is to redefine the paradigm of automated radiology report generation by revealing the relationship among the three sections in a report and how leveraging this relationship can benefit model training.
>
> Previous works like R2GenGPT treat radiology report generation as an image captioning task, taking X-ray images as the sole input to generate the findings and impression in a single step. However, this approach overlooks a critical aspect: the impression is an interpretation of the findings in the context of the clinical query provided in the indication, which is hard to derive from the image alone.
>
> Other works, such as MAIRA-1 and MAIRA-2, incorporate the indication and image as inputs but generate only the findings section. This deviates from real-world clinical practice, as it leaves the clinical query in the indication unanswered. In contrast, our work addresses these limitations by offering a practical framework that generates both the findings and impression sections guided by the indication to provide meaningful clinical insights.
>
> The second contribution of our work is the creation of a well-structured and cleaned dataset. As mentioned in the Introduction, the raw reports in MIMIC-CXR are in free-text form with sections that are not delineated. Our proposed version organizes the data into a clean and structured format, significantly enhancing its usability for developing radiology report generation methods with practical clinical applicability.
>
> **Q2**: Clarity: The authors concentrated on describing the 2-stage report generation algorithm, while the details of curating the MIMIC-1V3 dataset are not clearly stated. As the contribution of this paper heavily relies on the dataset creation, it is better to clearly address questions such as “How to handle reports without impression/finding/induction?”, “what algorithm was used to separate the report?” and “How to evaluate the correctness of the split?”.
>
> **A**: Section 4.1 explains the details of our three-fold effort to curate the MIMIC-1V3 dataset. The process involves heavy manual work to ensure the cleanness and structure of each report, such as rectifying misassigned sections.
>
> For preliminary report parsing, in line 113 of the Introduction, we mentioned that we adopted the MIMIC-CXR official code base (https://github.com/MIT-LCP/mimic-cxr). After automated parsing, we only retain the reports with all three sections. Then, we manually acquire a list of patterned sentences misassigned as part of the findings (these sentences mainly belong to the Technique and Comparison sections). Lastly, we use a simple string match algorithm to remove these patterned sentences from all reports.
>
> Regarding the split, as stated in paragraph 2 of Section 4.2, we use the official split file of MIMIC-CXR and make sure our training samples come only from the original training set and so on.

---

> ### Author Response · Authors · 2024-11-23
> **Response to Reviewer t2eU (2/2)**
>
> **Q3**: Please further highlight the technical contribution of the proposed RLRG framework, such as differences compared to existing multi-stage generation or MLLMs.
>
> **A**: As mentioned in Paragraph 3 of Related Works, there are two multi-stage generation frameworks, e.g., MedVersa and CheXagent. Both works do not leverage the indication as input during training. MedVersa takes the X-ray image as sole input and uses ChatGPT-generated template prompts to generate findings-only, impression-only, and complete reports (findings and impression). However, the authors of MedVersa have not granted us access to its pre-trained weight, which prevents us from evaluating their model. As for CheXagent, we have reported its results in Table 4.
>
> Other MLLM-based methods, such as MAIRA-1 and MAIRA-2 (closed-source), use the indication as input but only generate the findings section, which leaves the clinical query in the indication unanswered.
>
> To conclude, what differentiates our framework from previous works is that we redefine the paradigm of automated radiology report generation (RRG) from a clinical perspective, which narrows the gap between machine learning models and real-world bedside practice. Our training paradigm aims to verify the relationship among the indication, findings, and impression sections and how leveraging this relationship can benefit model performance.
>
> **Q4**: From an application of MLLM’s perspective, there is not much difference between indication + image → finding & impression and indication + image → finding → impression, as both findings and impressions are important and should be generated as medical reports. Please further explain the need to split the findings and impressions.
>
> **A**: Splitting the findings and the impression is necessary because generating the findings and generating the impression are inherently distinct tasks. From a clinical perspective, the findings section captures factual observations from the X-ray image, necessitating that the model focuses on learning accurate image-text alignment. In contrast, the impression section interprets the findings to address the clinical query posed in the indication, making it heavily reliant on the reasoning capabilities of the LLM. For instance, in Figure 1 (a), the impression states, `Findings suggestive of heart failure which could be secondary to volume overload,` directly addressing the clinical query `Eval for volume overload` in the indication. Specifically, the findings mention `pulmonary edema` and `mild cardiomegaly,` which are manifestations of `heart failure`. Thus, the impression is derived from reasoning over the findings.
>
> Attempting to generate both sections in a single step risks mixing these two distinct tasks, potentially reducing the efficiency of the training process. Moreover, separating the generation of findings and impressions ensures a more straightforward, structured final report with well-defined sections. As shown in Table 4, our method outperforms other LLM-based methods by a large margin .

---

> ### Comment · Reviewer_t2eU · 2024-11-28
> **Final Comment and Rating**
>
> My concerns still remain after the authors' rebuttal.
> First, I am not convinced that the proposed method is technically novel. Technically, the proposed framework is highly similar to previous works, except that the previous ones did not leverage/get access to the newly created dataset.
> Second, I think the contribution of this study is insufficient for a top-tier conference like ICLR. The key modification is to re-divide the existing dataset with different section settings, which is below the acceptance threshold despite the fact that it comes with good motivation.
> Therefore, I will maintain the original rating.

---

> > ### Author Response · Authors · 2024-11-29
> > **Response to Final Comment and Rating of Reviewer t2eU**
> >
> > Thank you for your constructive feedback and fair evaluation. We would like to clarify two things. First, the difference between our work and previous work is not the ability to leverage/get access to the newly created dataset. The difference is our profound understanding of the task of radiology report generation. Three sections are existing components of raw reports in the MIMIC-CXR dataset. However, previous works never realized the connections among each section nor utilized the connections to the full extent. Our framework proves: 1. The connections among the three sections do exist and can be leveraged to improve model performance. 2. Radiology report generation consists of two tasks: captioning and reasoning. Our works provides a clinically-aligned guideline for radiology report generation and is the first step towards generating clinically acceptable radiology report.
> >
> > Second, redividing the existing dataset is the first step in creating the MIMIC-1V3. Our dataset also underwent heavy manual correction for removing misassigned contents, which greatly improves its usability compared to MIMIC-CXR.

---

### Official Review · Reviewer_b8jm · 2024-11-03

**Soundness:** 3
**Presentation:** 3
**Contribution:** 2
**Rating:** 6
**Confidence:** 3

**Summary:**

The paper proposes the Radiologist-Like Progressive Generation (RLPG) framework for radiology report generation, which mirrors a radiologist's workflow by generating findings and then, based on these, producing the impression. To evaluate their approach, the paper introduce MIMIC-1V3, a dataset structured to match this workflow with separated indication, findings, and impression sections.

**Strengths:**

1. Easy to understand and intuitive Framework for report generation
2. Introducing a curated and refined public dataset for Chest X-Ray Report generation

**Weaknesses:**

1. The key experiment is missing in Table 3: Given the IMG and IND, generate the FIN and IMPR. This is critical because this is the authors' main claim: Separating the generation of each section is superior to the joint generation. Especially since in Table 2, the FIN only generation is inferior to the joint generation of FIN and IMPR on the test set.
2. In general, I would have expected more fair comparisons to other methods. For non of the approaches in Table 4 (except the authors approach) was the IND section provided. Simple experiments as providing the the IND section to the other pretrained models would be the minimum, especially since there are approaches such as MAIRA-1 [1], which also work with the IND section given.
3. RLPG sounds very much like a Chain-of-Thought (CoT) multi-stage approach. Comparing the approach to single-stage CoT could reveal interesting insights in how once could better guide the model in a single stage manner.

[1] Hyland, S. L., Bannur, S., Bouzid, K., Castro, D. C., Ranjit, M., Schwaighofer, A., ... & Alvarez-Valle, J. (2023). Maira-1: A specialised large multimodal model for radiology report generation. arXiv preprint arXiv:2311.13668.

[2] Wei, J., Wang, X., Schuurmans, D., Bosma, M., Xia, F., Chi, E., ... & Zhou, D. (2022). Chain-of-thought prompting elicits reasoning in large language models. Advances in neural information processing systems, 35, 24824-24837.

**Questions:**

Will the dataset and training code be made publicly available?
What is the reproducibility statement of the authors?

---

> ### Author Response · Authors · 2024-11-23
> **Response to Reviewer b8jm (1/3)**
>
> Dear Reviewer b8jm,
>
> Thank you for your detailed and constructive feedback on our work. We appreciate your insights into both the strengths and weaknesses of our approach, and we address each of your points below:
>
> **Weaknesses**
>
> **Q1**: The key experiment is missing in Table 3: Given the IMG and IND, generate the FIN and IMPR. This is critical because this is the authors' main claim: Separating the generation of each section is superior to the joint generation. Especially since in Table 2, the FIN only generation is inferior to the joint generation of FIN and IMPR on the test set.
>
> **A**: The table below shows the result of the experiment in which the model takes the image and the indication as input and generates the findings and the impression in one step (in the header, FIN indicates the evaluation results for findings,while IMPR for impression). Our approach still outperforms the one-step approach across all evaluation metrics on both findings and impression generation. This is because radiology report generation inherently consists of two distinct tasks: 1. findings generation (captioning): describe the factual observations of the radiograph that are relevant to the indication. 2. impression generation (reasoning): interpret the clinical implications of the findings and address the clinical query in the indication. Generating the findings and the impression in one step risks conflating these two tasks, which can lead to less accurate and coherent reports. By separating the generation process, our approach ensures that the factual observations are clearly and accurately detailed in the findings section, while the impression section provides a thoughtful and contextually relevant interpretation of these findings. This division not only enhances the clarity and reliability of the radiology reports but also aligns more closely with clinical workflows, where distinct attention is given to observations and their clinical implications. Consequently, our two-step method achieves superior performance across all evaluation metrics, demonstrating its effectiveness in producing high-quality, clinically useful radiology reports.
>
> In Table 2, the FIN-only generation achieved better scores on Natural Language Generation (NLG) metrics than the joint generation approach on the test set, and scores of the Clinical Efficacy (CE) are comparable. However, on the validation set, separated generation outperforms joint generation in both sections across all evaluation metrics. One potential reason is that checkpoint saving is only based on the scores of NLG metrics on the validation set during training. As we state in Section 4.2, there is a notable distribution shift between the validation and test sets, which could result in unstable results on the CE metrics.
>
> | Dataset  | Method |  Input | Output  | B-4 (FIN)  | CIDEr (FIN)  | Bert-S (FIN) | F1-all (FIN) |  RE-EM (FIN) |  RE-NLI (FIN) |  Rad-C (FIN) |B-4 (IMPR)|  CIDEr (IMPR)|  Bert-S (IMPR)|  F1-all (IMPR)|  RE-EM (IMPR) |  RE-NLI (IMPR) |  Rad-C (IMPR)|
> |---|---|---|---|---|---|---|---|---|---|---|---|---|---|---|---|---|---|
> |   | one-step  |  IMG+IND | FIN+IMPR  |  0.0654 | 0.1146  | 0.5426  | 0.3881  | 0.3498  | 0.2398  | 0.1971  | 0.0441  | 0.2895  | 0.3678  | 0.3543  | 0.2010  | 0.1397  |0.1101|
> | MIMIC-1V3  | two-step | IMG+IND  | FIN  | 0.1155  | 0.2856  | 0.5630  | 0.4516  |  0.3685 | 0.2430  | 0.2026  | -  | -  | -  | -  | -  | -  | -  |
> |   | two-step  | IMG+IND+gen_FIN  | IMPR  | -  | -  | -  | -  | -  | -  | -  | 0.0648  | 0.7547  | 0.4581  | 0.4308  | 0.2773  | 0.2206  | 0.1727  |

---

> ### Author Response · Authors · 2024-11-23
> **Response to Reviewer b8jm (2/3)**
>
> **Q2**: More fair comparisons to other methods with IND section provided.
>
> **A**: We appreciate your concerns regarding the fairness of the comparison. The indication is not external prior knowledge but an inherent part of a radiology report[1][2], and previous works fail to explore the connection among three sections. While MAIRA-1&2 work with the indication, they are closed-source, and only generate the findings without fully utilizing the indication, leaving clinical questions unanswered. In our original Table 4, we compared their results strictly under their original settings.
>
> To ensure a more fair comparison, we add the indication as input for other models in the following Table. Our approach still outperforms these LLM-based methods even adding the indication.
> For R2GenGPT, we add the indication to its original prompt and retrain the model.
> For the findings, we observe reduced scores for all evaluation metrics except the RE-NLI, which increases from 0.2363 to 0.2398. For the impression, adding indication can improve the model's performances on Bert-Score (0.3277 → 0.3678), RE-EM (0.1539 → 0.2010), and Rad-Complete (0.0869 → 0.1101), while negatively impact the performance on B-4 (0.0532 → 0.0441), CIDEr (0.4530 → 0.2895), F1-all (0.3875 → 0.3543), and RE-NLI (0.1657 → 0.1397).
>
> Directly adding the indication without crafting R2GenGPT's prompt increased the complexity and provided no explicit instructions for the model on how to utilize the extra information, which could explain the performance drops. Nevertheless, the clinical information in the indication is more relevant to the impression section, and this information likely contributed to the improvements in certain scores.
>
> For RadFM and CheXagent, we add the indication into their prompts and conduct inference on the MIMIC-1V3 test set. Adding the indication enhanced RadFM's performance in both sections, with increased scores on all evaluation metrics except for CIDEr and RE-NLI. RadFM's extensive pre-training enables the model to comprehend and utilize the information in the indication, highlighting the importance of the indication. However, since we performed inference without additional training, it is difficult to explain the observed decreases in the CIDEr and RE-NLI scores.
>
> In contrast, for CheXagent, adding the indication led to a significant decline in finding generation performance. Conversely, including the indication improved impression generation performance, with all evaluation metrics showing enhancements. We cannot perform a detailed analysis since CheXagent has not open-sourced its code and training data. Therefore, we can only conclude that including the indication benefits impression generation. This finding aligns with our conclusion that the indication and the impression are closely related.
>
> [1]. Evaluating the referring physician’s clinical history and indication as a means for communicating chronic conditions that are pertinent at the point of radiologic interpretation. J Digit Imaging 2015
>
> [2]. Good practice for radiological reporting. Guidelines from the European Society of Radiology (ESR)
>
> | Dataset  | Method |  Input | Output  | B-4 (FIN)  | CIDEr (FIN)  | Bert-S (FIN) | F1-all (FIN) |  RE-EM (FIN) |  RE-NLI (FIN) |  Rad-C (FIN) |B-4 (IMPR)|  CIDEr (IMPR)|  Bert-S (IMPR)|  F1-all (IMPR)|  RE-EM (IMPR) |  RE-NLI (IMPR) |  Rad-C (IMPR)|
> |-|-|-|-|-|-|-|-|-|-|-|-|-|-|-|-|-|-|
> |   | R2GenGPT  |  IMG | FIN+IMPR  |  0.1044 | 0.1530  | 0.5595  | 0.4476  | 0.3631  | 0.2363  | 0.1981  | 0.0532  | 0.4530  | 0.3277  | 0.3875  | 0.1539  | 0.1657  | 0.0869  |
> |   | R2GenGPT | IMG+IND  | FIN+IMPR  | 0.0654  | 0.1146  | 0.5426  | 0.3881  |  0.3498 | 0.2398  | 0.1971  | 0.0441  | 0.2895  | 0.3678  | 0.3543  | 0.2010  | 0.1397  | 0.1101  |
> |   | RadFM  | IMG  | FIN+IMPR  | 0.0003  | 0.0110  | 0.1413  | 0.1582  | 0.1005  | 0.0589  | 0.0434  | 0.0041  | 0.1783  | 0.1526  | 0.2423  | 0.0643  | 0.0512  | 0.0397  |
> |   | RadFM  | IMG+IND  | FIN+IMPR  | 0.0027  |  0.0060 | 0.2363  | 0.2422  | 0.1133  | 0.0463  | 0.0465  | 0.0089  | 0.0866  | 0.2109  | 0.2571  | 0.0911  | 0.0457  | 0.0445  |
> | MIMIC-1V3  | CheXagent  | IMG  | FIN  |  0.0540 | 0.0732  | 0.5152  | 0.2588  | 0.3484  | 0.2639  | 0.1740  | -  | -  | -  | -  | -  | -  | -  |
> |   | CheXagent  | IMG+IND  | FIN  |  0.0186 | 0.0431  | 0.4094  | 0.2216  |  0.2315 | 0.1694  | 0.1159  | -  | -  | -  | -  | -  |  - | -  |
> |   |  CheXagent | IMG  | IMPR  | -  | -  | -  | -  | -  | -  | -  | 0.0305  | 0.6147  | 0.3807  | 0.3749  | 0.2169  | 0.1753  | 0.1323  |
> |   |  CheXagent | IMG+IND  | IMPR  |  - |  - | -  | -  | -  | -  | -  |  0.0342 | 0.6374  | 0.4019  | 0.3811  | 0.2460  | 0.2072  | 0.1499  |
> |   | Ours  | IMG+IND  | FIN  | 0.1155  | 0.2856  | 0.5630  | 0.4516  | 0.3685  | 0.2430  | 0.2026  | -  | -  | -  | -  | -  | -  | -  |
> |   | Ours  | IMG+IND+gen_FIN | IMPR  | -  |  - | -  | -  | -  | -  | -  | 0.0648  | 0.7547  | 0.4581  | 0.4308  | 0.2773  | 0.2206  | 0.1727  |

---

> ### Author Response · Authors · 2024-11-23
> **Response to Reviewer b8jm (3/3)**
>
> **Q3**: RLPG sounds very much like a Chain-of-Thought (CoT) multi-stage approach. Comparing the approach to single-stage CoT could reveal interesting insights in how once could better guide the model in a single stage manner.
>
> **A**: We design the following prompt for one-stage CoT training.
>
> **Prompt**: `Given the patient's indication: <indication>, generate a findings section, which is the factual observation of the chest x-ray image. Then based on the indication and the generated findings, generate a diagnostic impression section which answers the clinical question in the indication.`
>
> The result is shown below. The drop in performance could be attributed to the increased complexity of the prompt, which introduces multiple sequential tasks that may overwhelm the model's processing capabilities. By requiring the generation of both a findings section and a diagnostic impression in a single prompt, the model faces a higher cognitive load, potentially leading to confusion or errors in task execution.
>
> | Dataset  | Method |  Input | Output  | B-4 (FIN)  | CIDEr (FIN)  | Bert-S (FIN) | F1-all (FIN) |  RE-EM (FIN) |  RE-NLI (FIN) |  Rad-C (FIN) |B-4 (IMPR)|  CIDEr (IMPR)|  Bert-S (IMPR)|  F1-all (IMPR)|  RE-EM (IMPR) |  RE-NLI (IMPR) |  Rad-C (IMPR)|
> |---|---|---|---|---|---|---|---|---|---|---|---|---|---|---|---|---|---|
> |   | one-step  |  IMG | FIN+IMPR  |  0.1044 | 0.1530  | 0.5595  | 0.4476  | 0.3631  | 0.2363  | 0.1981  | 0.0532  | 0.4530  | 0.3277  | 0.3875  | 0.1539  | 0.1657  |0.0869|
> | MIMIC-1V3  | one-step CoT| IMG+IND  | FIN+IMPR  | 0.0560  | 0.0825  | 0.4517  | 0.4183  |  0.3290 | 0.2295  | 0.1847  | 0.0361  | 0.2209  | 0.3463  | 0.3701  | 0.1884  | 0.1165  | 0.1083  |
> |  | Ours | IMG+IND  | FIN  | 0.1155  | 0.2856  | 0.5630  | 0.4516  |  0.3685 | 0.2430  | 0.2026  | -  | -  | -  | -  | -  | -  | -  |
> |   | Ours  | IMG+IND+gen_FIN  | IMPR  | -  | -  | -  | -  | -  | -  | -  | 0.0648  | 0.7547  | 0.4581  | 0.4308  | 0.2773  | 0.2206  | 0.1727  |
>
> **Q4**: Will the dataset and training code be made publicly available? What is the reproducibility statement of the authors?
>
> **A**: In section 5.1, we have provided adequate implementation details. We will release the source code and MIMIC-1V3 dataset upon acceptance.

---

> > ### Comment · Reviewer_b8jm · 2024-11-25
> >
> > Dear Authors,
> > Thank you for the comprehensive and detailed response. I believe including the additional experiments in the paper would further strengthen it and enhance its overall impact.
> >
> > My concerns have been addressed, and I am raising my score.

---

> > > ### Author Response · Authors · 2024-11-29
> > >
> > > Thank you for your thoughtful and encouraging feedback. We are grateful for your careful review and for raising your score. Your constructive comments have been invaluable in improving the quality and impact of our work.

---

### Official Review · Reviewer_XTQR · 2024-11-04

**Soundness:** 1
**Presentation:** 2
**Contribution:** 2
**Rating:** 3
**Confidence:** 5

**Summary:**

The authors introduce a Radiologist-Like Progressive Generation (RLPG) framework for report generation, which emulates the radiologist's workflow by first focusing on findings recognition through visual understanding, followed by diagnostic reasoning. They also present a new benchmark, MIMIC-1V3, derived from MIMIC-CXR, where each report is segmented into three sections—indication, findings, and impression—and paired with a corresponding frontal radiograph. The RLPG method is then compared with state-of-the-art LLM-based report generation models.

**Strengths:**

The newly developed MIMIC-1V3 benchmark serves as a standardized test bed for future report generation research.

**Weaknesses:**

1. The proposed framework may not fully align with real-world medical practices. Although MIMIC-CXR reports follow the order of indication, findings, and impression, clinicians may not always document in this structured sequence, which raises questions about the practical motivation for this approach.
2. The study lacks a detailed pipeline error analysis to assess how intermediate stages impact final output quality.
3. Comparisons in Table 4 may not be entirely fair, as the proposed model leverages the indication as prior knowledge in report generation.

**Questions:**

1. Does this approach assume a close relationship between indication and final diagnosis? Have the authors examined how indications relate to impressions or findings in the MIMIC-CXR dataset? Additionally, how does this approach perform if provided with an incorrect indication as input?
2. To ensure fair comparisons in Table 4, could the authors evaluate their model without using indication as input? Alternatively, if using indication as input, they could first generate the indication and report its intermediate results.
3. How does the clinical accuracy of findings influence the generation of the impression section?

---

> ### Author Response · Authors · 2024-11-23
> **Response to Reviewer XTQR (1/3)**
>
> Dear Reviewer XTQR,
>
> Thank you for your detailed and constructive feedback on our work. We appreciate your insights into both the strengths and weaknesses of our approach, and we address each of your points below:
>
> **Weaknesses**
>
> **Q1**: The proposed framework may not fully align with real-world medical practices. Although MIMIC-CXR reports follow the order of indication, findings, and impression, clinicians may not always document in this structured sequence, which raises questions about the practical motivation for this approach.
>
> **A**: We appreciate your concerns about the generalizability of our proposed framework. However, the indication, the findings, and the impression are essential for a radiology report. [1] states that indications provided with a radiological examination are critical components of a quality interpretation by the radiologist. [2] lists clinical referrals (which have the same function as indications, with different naming conventions), findings, and impression sections as elements of a report. The importance of these sections underscores the need for a framework that captures their distinct roles.
>
> Besides, although the raw reports in MIMIC-CXR have section keywords,
> without closely examining the functions of each section, the community tends to treat radiology report generation as an image captioning task. The free-text form of raw reports may exacerbate this misconception.
>
> Realizing the limitations of previous work, our work's practical motivation is to redefine the paradigm of automated radiology report generation from one-step image captioning to a two-step captioning-then-reasoning process. To facilitate this paradigm transition, our proposed dataset lays the foundation for bringing the radiology report generation task back to a state that closely aligns with medical practice.
>
> [1]. Evaluating the referring physician’s clinical history and indication as a means for communicating chronic conditions that are pertinent at the point of radiologic interpretation. J Digit Imaging 2015
>
> [2]. Good practice for radiological reporting. Guidelines from the European Society of Radiology (ESR)
>
> **Q2**: The study lacks a detailed pipeline error analysis to assess how intermediate stages impact final output quality. How does the clinical accuracy of findings influence the generation of the impression section?
>
> **A**: As is shown in Table 3, after replacing generated findings with ground-truth findings as input for impression generation, we observed elevated scores across all evaluation metrics. To mitigate the impact, we specifically include X-ray images as part of the inputs for impression generation, as image features could compensate for the missing or inaccurate information in the intermediate stage.
>
> We also conduct experiments that exclude X-ray images as input for impression generation. Since the inputs are pure text, we use Lora fine-tuning to add trainable parameters to the frozen LLM. The table below shows that replacing generated findings with ground-truth findings can improve the quality of impression generation for both methods. However, the magnitude of the improvement for different methods varies.
>
> We calculate the relative improvements after using ground truth findings as input, and for easy comparison, we calculate the mean of five clinical efficacy metrics. Without image features, ground truth findings increase the B-4 score from 0.0383 to 0.1428, an improvement of 0.1045. This represents a 273% increase relative to the original score of 0.0383. The increases in CIDEr score and average clinical efficacy (CE) score relative to their original scores are 228% and 68%, respectively. When including images as input, ground truth findings increase the B-4 score from 0.0684 to 0.2074, an improvement of 0.1426. This represents a 220% increase relative to the original score of 0.0684. The increases in CIDEr score and average CE score relative to their original scores are 121% and 55%, respectively. These results prove that including images as input can increase the robustness of our model, making it less sensitive to the quality of the findings.
>
> |  Dataset | Method  | Input  | Output  | B-4  | CIDEr  |  Bert-S | F1-all  | RE-EM  | RE-NLI  | Rad-C  | Avg-CE|
> |---|---|---|---|---|---|---|---|---|---|---|---|
> |   |  LoRA |  IND+gen_FIN | IMPR  |  0.0383 | 0.1166  | 0.3305  | 0.3719  |  0.2070 | 0.1338  | 0.1103  | 0.2307  |
> |   |  LoRA |  IND+gt_FIN |  IMPR | 0.1428 (273% &#8593;) | 0.3824 (228% &#8593;) |  0.5249 |  0.6042 | 0.3862  | 0.2308  | 0.1934  | 0.3879 (68% &#8593;) |
> | MIMIC-1V3  | Ours  | IND+IMG+gen_FIN  | IMPR  | 0.0648  | 0.7547  |  0.4581 | 0.4308  | 0.2773  | 0.2206  | 0.1727  | 0.3119  |
> |   |  Ours |  IND+IMG+gt_FIN |  IMPR | 0.2074 (220% &#8593;) |  1.6740 (121% &#8593;)| 0.5897  | 0.6614  | 0.4892  | 0.3513  | 0.3306  | 0.4844 (55% &#8593;) |

---

> ### Author Response · Authors · 2024-11-23
> **Response to Reviewer XTQR (2/3)**
>
> **Q3**: Comparisons in Table 4 may not be entirely fair, as the proposed model leverages the indication as prior knowledge in report generation.
>
> **A**: We appreciate your concerns regarding the fairness of the comparison. However, the indication is not external prior knowledge but an existing part of the MIMIC-CXR dataset. Moreover, the indication is a universal and critical component of the radiology report [1][2]. Previous works overlook the importance of indication and do not explore the connection among three sections during training. In our original Table 4, we compared their results strictly under their original settings.
>
> To address the concern and ensure a more fair comparison, we add the indication as input for other models in the following Table (FIN indicates the evaluation results are for Findings, while IMPR for impression). Our approach still outperforms these LLM-based methods even adding the indication. For R2GenGPT, we add the indication to its original prompt and retrain the model. For the findings section, we observe reduced scores for all evaluation metrics except the RE-NLI, which increases from 0.2363 to 0.2398. For the impression section, adding indication as input can improve the model's performances on Bert-Score (0.3277 → 0.3678), RE-EM (0.1539 → 0.2010), and Rad-Complete (0.0869 → 0.1101), while negatively impact the performance on B-4 (0.0532 → 0.0441), CIDEr (0.4530 → 0.2895), F1-all (0.3875 → 0.3543), and RE-NLI (0.1657 → 0.1397). Directly adding the indication without crafting R2GenGPT's prompt increased the complexity and provided no explicit instructions for the model on how to utilize the extra information, which could explain the performance drops. Nevertheless, the clinical information in the indication is more relevant to the impression section, and this additional information likely contributed to the improvements in certain scores.
>
> For RadFM and CheXagent, we added the indication into their prompts and conduct inference on the MIMIC-1V3 test set. Adding the indication enhanced RadFM's performance in generating both sections, with increased scores observed across all evaluation metrics except for CIDEr and RE-NLI in both sections. RadFM's extensive pre-training enables the model to comprehend and effectively utilize the information in the indication, highlighting the importance of the indication from a different perspective. However, since we performed inference without additional training, it is difficult to explain the decreases in the CIDEr and RE-NLI scores.
>
> In contrast, for CheXagent, adding the indication led to a significant decline in finding generation performance across all evaluation metrics; conversely, including the indication improved impression generation performance, with all evaluation metrics showing enhancements. We cannot perform a detailed analysis since CheXagent has not open-sourced its code and training data. Therefore, we can only conclude that including the indication benefits impression generation. This finding aligns with our conclusion that the indication and the impression are closely related.
>
> | Dataset  | Method |  Input | Output  | B-4 (FIN)  | CIDEr (FIN)  | Bert-S (FIN) | F1-all (FIN) |  RE-EM (FIN) |  RE-NLI (FIN) |  Rad-C (FIN) |B-4 (IMPR)|  CIDEr (IMPR)|  Bert-S (IMPR)|  F1-all (IMPR)|  RE-EM (IMPR) |  RE-NLI (IMPR) |  Rad-C (IMPR)|
> |-|-|-|-|-|-|-|-|-|-|-|-|-|-|-|-|-|-|
> |   | R2GenGPT  |  IMG | FIN+IMPR  |  0.1044 | 0.1530  | 0.5595  | 0.4476  | 0.3631  | 0.2363  | 0.1981  | 0.0532  | 0.4530  | 0.3277  | 0.3875  | 0.1539  | 0.1657  | 0.0869  |
> |   | R2GenGPT | IMG+IND  | FIN+IMPR  | 0.0654  | 0.1146  | 0.5426  | 0.3881  |  0.3498 | 0.2398  | 0.1971  | 0.0441  | 0.2895  | 0.3678  | 0.3543  | 0.2010  | 0.1397  | 0.1101  |
> |   | RadFM  | IMG  | FIN+IMPR  | 0.0003  | 0.0110  | 0.1413  | 0.1582  | 0.1005  | 0.0589  | 0.0434  | 0.0041  | 0.1783  | 0.1526  | 0.2423  | 0.0643  | 0.0512  | 0.0397  |
> |   | RadFM  | IMG+IND  | FIN+IMPR  | 0.0027  |  0.0060 | 0.2363  | 0.2422  | 0.1133  | 0.0463  | 0.0465  | 0.0089  | 0.0866  | 0.2109  | 0.2571  | 0.0911  | 0.0457  | 0.0445  |
> | MIMIC-1V3  | CheXagent  | IMG  | FIN  |  0.0540 | 0.0732  | 0.5152  | 0.2588  | 0.3484  | 0.2639  | 0.1740  | -  | -  | -  | -  | -  | -  | -  |
> |   | CheXagent  | IMG+IND  | FIN  |  0.0186 | 0.0431  | 0.4094  | 0.2216  |  0.2315 | 0.1694  | 0.1159  | -  | -  | -  | -  | -  |  - | -  |
> |   |  CheXagent | IMG  | IMPR  | -  | -  | -  | -  | -  | -  | -  | 0.0305  | 0.6147  | 0.3807  | 0.3749  | 0.2169  | 0.1753  | 0.1323  |
> |   |  CheXagent | IMG+IND  | IMPR  |  - |  - | -  | -  | -  | -  | -  |  0.0342 | 0.6374  | 0.4019  | 0.3811  | 0.2460  | 0.2072  | 0.1499  |
> |   | Ours  | IMG+IND  | FIN  | 0.1155  | 0.2856  | 0.5630  | 0.4516  | 0.3685  | 0.2430  | 0.2026  | -  | -  | -  | -  | -  | -  | -  |
> |   | Ours  | IMG+IND+gen_FIN | IMPR  | -  |  - | -  | -  | -  | -  | -  | 0.0648  | 0.7547  | 0.4581  | 0.4308  | 0.2773  | 0.2206  | 0.1727  |

---

> ### Author Response · Authors · 2024-11-23
> **Response to Reviewer XTQR (3/3)**
>
> **Q4**: Does this approach assume a close relationship between indication and final diagnosis? Have the authors examined how indications relate to impressions or findings in the MIMIC-CXR dataset? Additionally, how does this approach perform if provided with an incorrect indication as input?
>
> **A**: Our approach is rooted in the close relationship between the indication, the findings, and the impression. The relationship has been validated by [1,2,3].
>
> The indication contains the patient's brief medical history and a clinical question. The findings are the factual observation of the image. The impression section contains radiologists' diagnoses followed by the most supportive findings as evidence for making such a conclusion. These diagnoses are inferred from the findings based on the information in the indication. An example from the MIMIC-CXR dataset was shown in Figure 1 (a), where the impression states, `Findings suggestive of heart failure which could be secondary to volume overload,` which directly addresses the clinical query `Eval for volume overload` in the indication. To be more specific, the findings mentioned `pulmonary edema` and `mild cardiomegaly`, which are considered the manifestations of heart failure. In other words, `heart failure` is inferred from the findings.
>
> In real-world scenarios, providing an incorrect indication as input is considered malpractice, as mislabeling patient information could lead to serious diagnostic errors. Therefore, we do not see the necessity of evaluating the approach under such conditions, as this work focuses on improving performance with accurate and reliable inputs aligned with clinical practice.
>
> [3]. How to create a great radiology report. RadioGraphics, 2020.
>
> **Q5**: To ensure fair comparisons in Table 4, could the authors evaluate their model without using indication as input?
>
> **A**: As described in Q3, we have discussed incorporating the indication as input for other compared models. In Table 2, we have reported the results of our method without the indication as input. Compared to other methods in Table 4, without the indication as input, the impression section generated by our approach achieves the best scores across most evaluation metrics except for CIDEr and RE-NLI, which are only lower than CheXagent (0.6060 vs 0.6147 for CIDEr and 0.1686 vs 0.1753 for RE-NLI), and the findings section generated by our model achieves the best scores on Natural Language Generation (NLG) metrics and comparable clinical efficacy (CE) scores.
>
> **Q6**: Alternatively, if using indication as input, they could first generate the indication and report its intermediate results.
>
> **A**: We believe it is unnecessary to generate the indication if it is already provided as input, as this would introduce redundancy rather than contribute to the overall process. Additionally, indications are typically supplied by the ordering physician and are not generated or written by radiologists. Therefore, generating indications would not align with the clinical workflow.

---

> > ### Comment · Reviewer_XTQR · 2024-11-26
> >
> > 1. The authors did not directly address my question in Q1. I did not imply that the "indication" is not part of the radiology report. My question concerns the order of writing the indication, findings, and impression, which may not align with clinical practice. Writing the indication could require a similar effort as writing the findings or impression. Therefore, the proposed method, which uses the indication as prior knowledge, may not significantly reduce the radiologist's workload.
> >
> > 2. The pipeline error analysis should assess the impact of the indication, as it plays a critical role in the proposed method.
> >
> > 3. While previous methods relied solely on images for radiology report generation, the proposed method incorporates the indication as additional input, treating it as prior knowledge. I find the authors' argument that this is not external prior knowledge—since it is part of the MIMIC-CXR dataset—confusing. Additionally, adding the indication to the prompts of RadFM and CheXAgent might create a somewhat fairer comparison to the previous experimental setup. However, the comparison remains unfair because the proposed method uses the indication for training, while the others do not.
> >
> > 4. Have the authors attempted to predict or generate the indication? If so, what is the performance? If accurately generating the indication proves to be as challenging as generating findings and impressions, this would support my argument that using the indication as input might not reduce the radiologist’s workload. If not, the authors should use generated indications as input and perform a fair comparison with RadFM and CheXAgent.
> >
> > 5. In response to Q5, the authors claim that without using the indication as input, the proposed approach achieves the highest scores for the impression section across most evaluation metrics, and the findings section achieves the best scores on NLG metrics and comparable CE scores. However, RadFM and CheXAgent were trained on larger datasets. I am curious how the proposed method outperforms these approaches. Could the authors provide more details about the experiments? Was the proposed model trained without the indication, or did it simply not use the indication as input while being trained with it?

---

> > > ### Author Response · Authors · 2024-11-29
> > > **Response to Reviewer XTQR (1/2)**
> > >
> > > Dear reviewer XTQR: Thank you for your thorough and constructive comments. We appreciate your efforts in reviewing the details of our work. We will address your new questions in the following section.
> > >
> > > **Response to Q1**: First, we want to clarify that the indication is `NOT` written by the radiologist. Instead, the indication--which encompasses the patient's brief medical history and the clinical query--is provided by the `referring physician` and serves as the primary reason for ordering the radiological examination. Therefore, incorporating the indication as prior knowledge does not increase the radiologist's workload.
> > >
> > > According to [1], in today's health care system, the clinical history and `indication` (CHI) is oftentimes the only information `provided by the referring physician` to the radiologist. In addition, [2] states that clinical referral (which has the same function as the indication, with different naming conventions) is of the utmost importance that the clinical history of the patient is `provided by the referring physician`, to enable correct image interpretation.
> > >
> > > In other words, the indication exists before the X-ray image is acquired and before the radiologist writes the findings and impression.
> > >
> > > The sequential order of the indication, findings, and impression in a radiology report closely aligns with real-world clinical practice. [3] states, "The typical radiology report follows the logical and inductive structure of a description of the findings followed by a discussion of the differential diagnosis and a conclusion ...The American College of Radiology handbook for
> > > residents divides the radiology report into six sections:
> > > examination, history/indication, technique, comparison, findings, and impression." Moreover, [2] also follows the indication, findings, and impression order. This order is confirmed by the American College of Radiology and the European Society of Radiology.
> > >
> > > To sum up, the findings contain the factual observations of the X-ray image, and the impression interprets the clinical implications of the findings by giving a diagnosis. Placing the impression after the findings is logical and strictly aligns with clinical practice.
> > >
> > > [1]. Obara, Piotr, et al. "Evaluating the referring physician's clinical history and indication as a means for communicating chronic conditions that are pertinent at the point of radiologic interpretation." Journal of digital imaging 28 (2015): 272-282.
> > >
> > > [2]. European Society of Radiology (ESR) http://www.myESR.org communications@myESR.org. "Good practice for radiological reporting. Guidelines from the European Society of Radiology (ESR)." Insights into imaging 2.2 (2011): 93-96.
> > >
> > > [3]. Wallis, A., and P. McCoubrie. "The radiology report--are we getting the message across?." Clinical radiology 66.11 (2011): 1015-1022.
> > >
> > > **Response to Q2**: In Table 3 of the main paper, we have compared the performance of my model with and without the indication as input on both the test set and validation set. In section 5.3, four paragraphs are dedicated to discussing the benefits of using indication as input.
> > >
> > > **Response to Q3**: While the indication is part of the MIMIC-CXR dataset, it is not written by the radiologists as part of the radiology report writing process. Instead, as explained in our response to Q1, it is provided by the referring physician.
> > >
> > > Achieving absolute fairness in model comparisons is inherently challenging due to multiple factors that can influence each model's performance and outcomes. Ideally, the fairest approach would involve retraining competing models from scratch using the MIMIC-1V3 dataset.
> > >
> > > However, this ideal scenario is impractical for several reasons. First, CheXagent has not open-sourced its code, which impedes our ability to retrain the model using the MIMIC-1V3 dataset. Second, RadFM utilizes MedLLaMA-13B, which is substantially larger than our LLM, LLaMA2-7B. The size difference between the LLMs makes a fair comparison even harder. Moreover, both CheXagent and RadFM involve fully fine-tuning their respective LLMs, while our LLM remains frozen during training. The volume of MIMIC-1V3 is relatively low, which raises the concern that whether it is inadequate or not for fully fine-tuning the LLM.
> > > In summary, our current approach, adding the indication to the prompts of CheXagent and RadFM for direct inference, is more practical and reasonable.
> > >
> > > In addition, previous works have not been restricted from using the indication as input during training. Previous works did not utilize the indication because they overlooked its importance and misconceived the nature of automated radiology report generation as an image captioning task, which should be a captioning-then-reasoning task.
> > > We believe the seemingly "unfair" part of the comparison is actually the breakthrough of our work: redefining the task to better align with real-world clinical workflows by incorporating the indication to enhance reasoning and interpretation.

---

> > > ### Author Response · Authors · 2024-11-29
> > > **Response to Reviewer XTQR (2/2)**
> > >
> > > **Response to Q4**:  We have never considered indication prediction or generation. As mentioned in the response to Q1, the referring physician provides the indication. Roughly speaking, the indication is the X-ray image's background information, restricting the image's interpretation in a specific context. The indication is required by the radiologists to make accurate diagnoses, which should naturally be the input of deep learning models.
> > >
> > > To give more concrete examples, we select the following cases from the training set of MIMIC-1V3: {Study ID: 51588704, indication: M with HIV and cough. Eval pneumonia.} and {Study ID:50147677, indication: -year-old man status post transplant with cough for two weeks. Evaluate for pneumonia.}. Information such as `with HIV` or `status post transplant` cannot be directly inferred from the X-ray image. However, it alerts the radiologist to pay attention to certain findings as patients with HIV or transplant patients are prone to opportunistic pneumonia pathogens, which often present with bilateral interstitial infiltrates or a "ground-glass" appearance on imaging.
> > >
> > > **Response to Q5**: "without using the indication as input" means without the indication in both training and inference stages. In this case, our approach still achieves competitive results compared to RadFM and CheXagent.
> > >
> > > We want to assert that the evaluation results of CheXagent and RadFM on the MIMIC-1V3 test set are reliable. In the original CheXagent paper for the MIMIC-CXR evaluation, the authors report three evaluation metrics for findings generation: Bert-Score (0.504), CheXbert-Score (0.249), and RadGraph-Score (0.186), and only the Rouge-L (0.403) for impression generation. By rerunning their model on the MIMIC-1V3 test set, we obtain results for findings generation: Bert-Score (0.5152), CheXbert-Score (0.2588), and RadGraph-Score (0.1740) and Rouge-L (0.3747) for impression generation. These similar evaluation results confirm that we have correctly implemented their model.
> > >
> > > Conversely, the authors of RadFM provided 35 prompts (15 caption prompts and 20 report prompts) for generating radiology reports. Their model is highly sensitive to prompt variations, as discussed on their GitHub page. Our main paper used a caption prompt: `Can you provide a caption consisting of findings and impression for this medical image?'` (denoted as old_p). We reran their models using two additional report prompts: `Can you provide a radiology report for this medical image?` (p1) and `What are the findings presented in this medical scan?` (p2). The evaluation results are shown in the table below. We observe elevated scores across all metrics for p1 and p2 compared to old_p. So far, we have not identified a perfect prompt for RadFM in radiology report generation. Based on the available results, our approach still outperforms RadFM.
> > >
> > > Our approach outperforms CheXagent and RadFM in the report generation task because, different from their models, which are designed as multitask foundation models, ours is specialized for a single objective. Scaling up the visual instruction tuning data to the millions level empowers their models with excellent generalization ability. For instance, RadFM can process 2D and 3D images across multiple anatomical locations, while CheXagent can perform over 30 diverse tasks, such as Visual Question Answering (VQA), view classification, image-text matching, etc. While these multitask foundation models offer broad functionality, enhancing their generalization sometimes compromises specialized performance on a certain task--a trade-off that has been documented in general domain foundation models[4][5].
> > >
> > > [4]. InstructBLIP: Towards General-purpose Vision-Language Models with Instruction Tuning
> > >
> > > [5]. Liu, Haotian, et al. "Visual instruction tuning." Advances in neural information processing systems 36 (2024).
> > >
> > > | Dataset  | Method |  Input | Output  | B-4 (FIN)  | CIDEr (FIN)  | Bert-S (FIN) | F1-all (FIN) |  RE-EM (FIN) |  RE-NLI (FIN) |  Rad-C (FIN) |B-4 (IMPR)|  CIDEr (IMPR)|  Bert-S (IMPR)|  F1-all (IMPR)|  RE-EM (IMPR) |  RE-NLI (IMPR) |  Rad-C (IMPR)|
> > > |-|-|-|-|-|-|-|-|-|-|-|-|-|-|-|-|-|-|
> > > |   | RadFM (p1)  |  IMG | FIN+IMPR  |  0.0233 |  0.0557 | 0.3906  | 0.2182  |  0.2302 | 0.1759  | 0.1192  | 0.0155  | 0.5081  | 0.2113  |  0.2665 |  0.1298 | 0.1437  | 0.0855  |
> > > |   | RadFM (p2) | IMG  | FIN+IMPR  | 0.0236  | 0.0576  | 0.3917  | 0.2060  | 0.2381  | 0.1846  | 0.1224  | 0.0164  | 0.5072  | 0.2289  | 0.2802  | 0.1344  | 0.1431  | 0.0893  |
> > > |   | RadFM (old_p) | IMG  | FIN+IMPR  | 0.0003  | 0.0110  | 0.1413  | 0.1582  | 0.1005  | 0.0589  | 0.0434  | 0.0041  | 0.1783  | 0.1526  | 0.2423  | 0.0643  | 0.0512  | 0.0397  |
> > > |   | Ours  | IMG+IND  | FIN  | 0.1155  | 0.2856  | 0.5630  | 0.4516  | 0.3685  | 0.2430  | 0.2026  | -  | -  | -  | -  | -  | -  | -  |
> > > |   | Ours  | IMG+IND+gen_FIN | IMPR  | -  |  - | -  | -  | -  | -  | -  | 0.0648  | 0.7547  | 0.4581  | 0.4308  | 0.2773  | 0.2206  | 0.1727  |

---

> > > > ### Comment · Reviewer_XTQR · 2024-11-29
> > > >
> > > > 1. Pipeline error analysis is distinct from ablation studies, as it focuses on how upstream errors affect downstream performance. In this context, I am particularly interested in understanding how the accuracy of indications impacts the generation of radiology reports. Therefore, I recommend that the authors introduce distortions or noise to the indications and report the downstream performance at various noise levels. Table 3 does not adequately address my concerns regarding the impact of indication errors. Additionally, pipeline error analysis should also be applied to the process of generating impressions from findings.
> > > >
> > > > 2. The authors assert that their use of indications represents a breakthrough in their work; however, there is limited discussion on how these indications influence radiology report generation. As previously requested in my initial review, a thorough pipeline error analysis is necessary. Furthermore, the paper lacks statistical analysis examining the correlation between indications and impressions or findings. I suggest that the authors allocate more space in their manuscript to comprehensively discuss indications rather than relying solely on qualitative analyses and examples.
> > > >
> > > > 3. Moreover, I concur with Reviewer t2eU that utilizing indications does not constitute a methodological breakthrough. The proposed method appears instinctive rather than technically novel. This paper may be better suited for conferences such as MICCAI or medical-related journals.
> > > >
> > > > 4. The authors claim that CheXagent and RadFM are designed as multitask foundation models while theirs is specialized for a single objective, suggesting that multitask foundation models compromise specialized performance for specific tasks. Given this distinction, why did they not compare their methods with those specifically designed for radiology report generation, such as METransformer[1], KiUT[2], DCL[3], PromptMRG[4]?
> > > >
> > > > [1] Wang, Zhanyu, Lingqiao Liu, Lei Wang, and Luping Zhou. "Metransformer: Radiology report generation by transformer with multiple learnable expert tokens." In Proceedings of the IEEE/CVF Conference on Computer Vision and Pattern Recognition, pp. 11558-11567. 2023.
> > > >
> > > > [2] Huang, Zhongzhen, Xiaofan Zhang, and Shaoting Zhang. "Kiut: Knowledge-injected u-transformer for radiology report generation." In Proceedings of the IEEE/CVF Conference on Computer Vision and Pattern Recognition, pp. 19809-19818. 2023.
> > > >
> > > > [3] Li, Mingjie, Bingqian Lin, Zicong Chen, Haokun Lin, Xiaodan Liang, and Xiaojun Chang. "Dynamic graph enhanced contrastive learning for chest x-ray report generation." In Proceedings of the IEEE/CVF Conference on Computer Vision and Pattern Recognition, pp. 3334-3343. 2023.
> > > >
> > > > [4] Jin, Haibo, Haoxuan Che, Yi Lin, and Hao Chen. "Promptmrg: Diagnosis-driven prompts for medical report generation." In Proceedings of the AAAI Conference on Artificial Intelligence, vol. 38, no. 3, pp. 2607-2615. 2024.

---

> > > > > ### Author Response · Authors · 2024-11-29
> > > > >
> > > > > Thank you for your detailed comments. We would like to emphasize the following points: 1. ICLR does not exclude medical AI research from its relevant topics. Therefore, we deem our submission to ICLR is justifiable. In addition, our proposed method is not ``instinctive'' but based on established medical literature. 2. As shown in our submission's profile, the primary area we choose is datasets and benchmarks. Our work aims to redefine the automated radiology report generation paradigm and provide a clean and standardized benchmark to facilitate such a transition. Our framework, which closely aligns with radiologists' workflow, is designed to utilize the new benchmark. Our experiments support our claim that leveraging the relationships among the indication, findings, and impression can greatly benefit model performance.

---

> > > > > > ### Comment · Reviewer_XTQR · 2024-11-30
> > > > > > **Final Comment and Rating**
> > > > > >
> > > > > > My concerns remain and have even intensified. My evaluation has always been grounded in the quality of the paper, without bias due to its medical context. Overall, the paper lacks a comprehensive analysis of the curated dataset, such as statistical examinations of the correlation between indications and impressions or findings. The experimental results are insufficient to support the conclusions drawn, as the impact of indication errors is not adequately addressed. Furthermore, the baseline methods used for comparison are multitask foundation models, while the proposed method is tailored for a single objective. As the authors claim that multitask foundation models compromise specialized performance for specific tasks, they fail to compare their approach with state-of-the-art specialized methods. For these reasons, I will maintain my original rating.

---

> > > > > > > ### Author Response · Authors · 2024-12-02
> > > > > > > **Response to Final Comment and Rating of Reviewer XTQR**
> > > > > > >
> > > > > > > **Response**: We appreciate your time and effort in reviewing our work. Regarding your concerns in the final comment, we would like to respond with the following:
> > > > > > >
> > > > > > > `Statistical Analysis`: We employ Bio_ClinicalBERT, a domain-specific variant of BERT tailored for clinical text, to process the indication, findings, and impression, then extract the CLS token as aggregated information from the entire input sequence for calculating the cosine similarity of $indication \leftrightarrow findings$, $indication \leftrightarrow impression$, $findings \leftrightarrow impression$. The three types of similarity scores for the training, validation, test sets, and all combined are shown in the table below. There is moderate similarity between the indication and findings sections, indicating they share a reasonable degree of semantic overlap. The slight variations across splits are minimal, suggesting consistency. The high similarity between the indication and impression sections indicates a strong alignment, suggesting that the final impressions closely reflect the initial indications. The very high similarity of the findings and impression sections indicates they are almost identical in their semantic content. This suggests that impressions are directly derived from findings with minimal divergence.
> > > > > > > The **correlation** between $indication \leftrightarrow findings$\_similarity and $indication \leftrightarrow impression$\_similarity is **0.71**, indicating a strong positive correction. To sum up, the relationship between the indication, findings, and impression is not only based on the established medical literature but also statistically robust.
> > > > > > >
> > > > > > > | **Split**     | **$IND \leftrightarrow FIN$** | **$IND \leftrightarrow IMP$** | **$FIN \leftrightarrow IMP$** |
> > > > > > > |---------------|-------------------------------|-------------------------------|-------------------------------|
> > > > > > > | **train**     | $0.635 \pm 0.119$            | $0.699 \pm 0.100$            | $0.834 \pm 0.061$            |
> > > > > > > | **validation**| $0.673 \pm 0.122$            | $0.722 \pm 0.096$            | $0.835 \pm 0.064$            |
> > > > > > > | **test**      | $0.625 \pm 0.138$            | $0.699 \pm 0.113$            | $0.850 \pm 0.065$            |
> > > > > > > | **all**       | $0.635 \pm 0.120$            | $0.699 \pm 0.100$            | $0.834 \pm 0.061$            |
> > > > > > >
> > > > > > > `Indication Errors`: Defining and identifying indication errors poses a significant challenge, primarily because indications are provided by referring physicians, and we have no access to the patient's original electronic health records (EHR). In addition, the clinical significance of feeding the variations of the indication is unclear since ensuring the accuracy and consistency of input data is paramount in medical AI research, as it directly affects the reliability and validity of the models.
> > > > > > >
> > > > > > > `Comparison with other models`: Assessing the performance of our approach against existing task-specific models may lead to unfair comparisons for several reasons. Firstly, our model integrates a large language model (LLM) as the language decoder, resulting in a substantially larger architecture than the other models. This inherent difference in model complexity can skew performance metrics, making direct comparisons less meaningful. Secondly, models such as Metransformer and Kiut have not publicly released their code. Furthermore, most task-specific models are designed to process only image data. Incorporating textual information like indications necessitates significant modifications, such as adding a text encoder. The interactions and integrations between these newly added modules and the original architectures of these models remain unclear. For the reasons above, our current comparison strategy is reasonable and practical.

---

### Official Review · Reviewer_Y732 · 2024-11-04

**Soundness:** 2
**Presentation:** 3
**Contribution:** 2
**Rating:** 6
**Confidence:** 3

**Summary:**

The paper introduces the Radiologist-Like Progressive Generation (RLPG) framework, which mimics a radiologist's workflow for generating structured radiology reports. By sequentially generating findings and impression sections from a single input image and clinical indication, the model better aligns with clinical needs. The authors also introduce MIMIC-1V3, a dataset with structured radiology reports in distinct sections, facilitating focused benchmarking of report generation.

**Strengths:**

1. Alignment with Clinical Workflow: The RLPG framework emulates the actual workflow of radiologists, enhancing clinical relevance and report consistency, making it practical in real-world settings.
2. Dataset Curation: MIMIC-1V3 addresses inconsistencies in the original MIMIC-CXR dataset, providing a cleaner and more reliable benchmark for radiology report generation.

**Weaknesses:**

The RLPG framework seems straightforward and intuitive, especially in its logical progression and handling of radiology reports. Moreover,  aside from experimental metrics, there seems to be insufficient evidence demonstrating the innovation and superiority of this training paradigm.

**Questions:**

1. If the findings generated in the first stage contain inaccuracies, how does this affect the quality of the impression? Is there a mechanism to mitigate potential error propagation between stages?
2. Is there often overlapping information between the findings and impression sections? If so, does the model tend to produce redundant content, and could this result in outputs similar to previous frameworks?

---

> ### Author Response · Authors · 2024-11-23
> **Response to Reviewer Y732 (1/2)**
>
> Dear Reviewer Y732,
>
> Thank you for your detailed and constructive feedback on our work. We appreciate your insights into both the strengths and weaknesses of our approach, and we address each of your points below:
>
> **Weakness**
>
> **Q1**: The RLPG framework seems straightforward and intuitive, especially in its logical progression and handling of radiology reports. Moreover, aside from experimental metrics, there seems to be insufficient evidence demonstrating the innovation and superiority of this training paradigm.
>
> **A**:
> **Innovation**: Although our framework seems straightforward and intuitive, our breakthrough is to redefine the paradigm of automated radiology report generation (RRG) from a clinical perspective. Based on the established medical literature[1][2][3], we point out that the creation of a radiology report consists of two distinct tasks, e.g., 1. findings generation (captioning): describe the factual observations of the radiograph that are relevant to the indication. 2. impression generation (reasoning): interpret the clinical implications of the findings and address the clinical query in the indication. In contrast, previous works mainly treat RRG as a simple image captioning task, with no involvement or insufficient usage of the indication. Therefore, our training paradigm is not only closely aligned with real-world clinical practice but also the first work to use the indication to generate the findings and impression sections.
>
> Moreover, to better utilize the relationship among the three sections, we create a well-structured and cleaned dataset, MIMIC-1V3. As mentioned in the Introduction, the raw reports in MIMIC-CXR are in free-text form with sections that are not delineated. Our proposed version organizes the data into a clean and structured format, significantly enhancing its usability for developing radiology report generation methods with practical clinical applicability.
>
> [1]. Evaluating the referring physician’s clinical history and indication as a means for communicating chronic conditions that are pertinent at the point of radiologic interpretation. J Digit Imaging 2015
>
> [2]. Good practice for radiological reporting. Guidelines from the European Society of Radiology (ESR)
>
> [3]. How to create a great radiology report. RadioGraphics, 2020.
>
> **Performance**: To better demonstrate the superiority of our training paradigm, we have updated the Appendix section with generated reports of our framework and other LLM-based models. Please refer to section A.3 for more details.
>
> **Question**
>
> **Q2**: If the findings generated in the first stage contain inaccuracies, how does this affect the quality of the impression?
>
> **A**: Inaccurate findings would negatively impact the impression quality. This trend has been revealed in our experiments. As shown in Table 3, after replacing generated findings with ground-truth findings as input for impression generation, we observed elevated scores across all evaluation metrics. Given the inaccuracies that occurred in the first stage, our results still outperform end-to-end training and other LLM-based methods.

---

> ### Author Response · Authors · 2024-11-23
> **Response to Reviewer Y732 (2/2)**
>
> **Q3**: Is there a mechanism to mitigate potential error propagation between stages?
>
> **A**: Yes, we incorporate X-ray images as inputs for the second-stage impression generation to reduce potential error propagation between the two stages. Findings are the factual observation, or in simpler terms, the caption, of the X-ray image. Including X-ray images as input should compensate for the missing or inaccurate information in the findings. To demonstrate the quantitative effect of this mechanism, we also conduct experiments that exclude X-ray images as input for impression generation. Since the inputs are pure text, we use Lora fine-tuning to add trainable parameters to the frozen LLM. The table below shows that replacing generated findings with ground-truth findings can improve the quality of impression generation for both methods. However, the magnitude of the improvement for different methods varies.
>
> We calculate the relative improvements after using ground truth findings as input, and for easy comparison, we calculate the mean of five clinical efficacy metrics. Without image features, ground truth findings can increase the B-4 score of the impression from 0.0383 to 0.1428, an improvement of 0.1045. This represents a 273% increase relative to the original score of 0.0383. The increases in CIDEr score and average clinical efficacy (CE) score of the impression relative to their original scores are 228% and 68%, respectively. When including images as input, ground truth findings increase the B-4 score of the impression from 0.0684 to 0.2074, an improvement of 0.1426. This represents a 220% increase relative to the original score of 0.0684. The increases in CIDEr score and average CE score of the impression relative to their original scores are 121% and 55%, respectively. These results prove that including images as input for impression generation can increase the robustness of the model, and image features can compensate for the missing or inaccurate information in the findings.
>
> |  Dataset | Method  | Input  | Output  | B-4  | CIDEr  |  Bert-S | F1-all  | RE-EM  | RE-NLI  | Rad-C  | Avg-CE|
> |---|---|---|---|---|---|---|---|---|---|---|---|
> |   |  LoRA |  IND+gen_FIN | IMPR  |  0.0383 | 0.1166  | 0.3305  | 0.3719  |  0.2070 | 0.1338  | 0.1103  | 0.2307  |
> |   |  LoRA |  IND+gt_FIN |  IMPR | 0.1428 (273% &#8593;) | 0.3824 (228% &#8593;) |  0.5249 |  0.6042 | 0.3862  | 0.2308  | 0.1934  | 0.3879 (68% &#8593;) |
> | MIMIC-1V3  | Ours  | IND+IMG+gen_FIN  | IMPR  | 0.0648  | 0.7547  |  0.4581 | 0.4308  | 0.2773  | 0.2206  | 0.1727  | 0.3119  |
> |   |  Ours |  IND+IMG+gt_FIN |  IMPR | 0.2074 (220% &#8593;) |  1.6740 (121% &#8593;)| 0.5897  | 0.6614  | 0.4892  | 0.3513  | 0.3306  | 0.4844 (55% &#8593;) |
>
> **Q4**: Is there often overlapping information between the findings and impression sections? If so, does the model tend to produce redundant content, and could this result in outputs similar to previous frameworks?
>
> **A**: Yes, sometimes there is overlapping information between the findings and impression sections. But it is not redundant. From a clinical perspective, the impression section contains radiologists' diagnoses followed by the most supportive findings as evidence for making such a conclusion. These diagnoses are inferred from the findings based on the indication. For example, in Figure 1 (a), the impression states, `Findings suggestive of heart failure which could be secondary to volume overload,` directly addresses the indication's clinical query, `Eval for volume overload`. To be more specific, the findings mentioned `pulmonary edema` and `mild cardiomegaly`, which are considered the manifestations of `heart failure`. In other words, `heart failure` is inferred from the findings. To conclude, the overlapping part serves different purposes, i.e., the factual observation in the findings and the rationale for making diagnoses in the impression.
>
> The outputs of our model are certainly not similar to previous works for two reasons. First, previous works mainly combine the findings and the impression as the training target, which makes distinguishing the findings from the impression in the outputs challenging. In contrast, stage one of our model only uses the findings as the training target, and stage two only uses the impression as the training target. Second, since we incorporate the indication as input for both stages, the outputs of our model are more closely aligned with the clinical information in the indication. In particular stage two is specifically designed to address the clinical query presented in the indication. This design is the defining feature of our model, one that all previous works have overlooked.

---

> > ### Comment · Reviewer_Y732 · 2024-11-27
> >
> > Thank you to the authors for the rebuttal and the detailed explanations regarding the motivation and experimental results. Most of my concerns have been resolved, and I will increase my score to 6.

---

> > > ### Author Response · Authors · 2024-11-29
> > >
> > > Thank you for taking the time to carefully review our rebuttal and for your thoughtful feedback. We are pleased to hear that our explanations addressed most of your concerns and that you found our motivation and experimental results to be clarified. We sincerely appreciate your decision to raise your score and your recognition of our efforts to improve the clarity and impact of the paper.

---

### Meta-Review · Area_Chair_uyJ7 · 2024-12-22

**Metareview:**

The paper introduces the Radiologist-Like Progressive Generation (RLPG) framework, a multi-stage approach that emulates the radiologist's workflow for generating structured radiology reports by sequentially producing findings and impressions, evaluated on the newly curated MIMIC-1V3 dataset with segmented report sections.

Reviewers found this paper strong in its clinically relevant RLPG framework, which mirrors radiologists' workflows, and its curated MIMIC-1V3 dataset, providing a reliable benchmark for radiology report generation. Reviewers raised concerns about the RLPG framework's limited originality, resembling existing multi-stage approaches without sufficient evidence of its superiority (Reviewers Y732, t2eU). The practical alignment of RLPG with real-world medical workflows was questioned, as clinicians may not always document in a structured sequence (Reviewer XTQR). Additionally, the lack of critical experiments to validate the claimed benefits of section-wise generation, insufficiently fair comparisons with other methods that could utilize the indication section, and the absence of detailed pipeline error analyses limit the study's ability to demonstrate the robustness and effectiveness of its approach (Reviewers b8jm, XTQR). Additionally, unclear curation details for MIMIC-1V3 undermine the dataset's validity and reproducibility (Reviewer t2eU).

During the discussion, the authors emphasized RLPG's originality in redefining radiology report generation through its clinically aligned two-stage paradigm and the creation of the structured MIMIC-1V3 dataset, distinguishing it from existing approaches. They conducted additional experiments to validate the benefits of section-wise generation and incorporating the indication section for comparison with other methods. However, they acknowledged the limitations in performing detailed pipeline error analyses and retraining closed-source models. The authors provided additional details about the curation process for MIMIC-1V3. However, some aspects, such as the evaluation of the correctness of the split and handling of missing sections, remain insufficiently detailed. The reviewers' final feedback was mixed. Reviewer Y732 and b8jm raised their scores, citing resolved concerns and strong responses, while XTQR and t2eU maintained their original ratings, citing unresolved issues with dataset analysis, insufficient comparisons, weak experimental support, and limited technical novelty.

After reviewing the paper, the reviewers' comments, the authors’ rebuttal, and discussions, it is evident that significant concerns remain. While the paper introduces a clinically motivated framework and a curated dataset for radiology report generation, critical issues persist. The dataset, a key claimed contribution, lacks sufficient details on its curation process and statistical validation, raising questions about its validity and reproducibility. Additionally, the technical contribution is incremental, offering limited novelty compared to existing multi-stage and specialized methods. The notable feature appears to be the inclusion of the "indication" section as input, which, however, is not universally available in datasets like IU-Xray. Furthermore, the work lacks comprehensive comparisons with state-of-the-art radiology report generation methods, particularly recent advancements from 2024, limiting its contextual positioning within the field. Despite the authors’ responses and partial acknowledgment of the dataset's potential by reviewers, these unresolved concerns constrain the work's readiness for publication.

**Additional Comments On Reviewer Discussion:**

During the discussion, the authors emphasized RLPG's originality in redefining radiology report generation through its clinically aligned two-stage paradigm and the creation of the structured MIMIC-1V3 dataset, distinguishing it from existing approaches. They conducted additional experiments to validate the benefits of section-wise generation and incorporate the indication section for comparison with other methods. However, they acknowledged the limitations in performing detailed pipeline error analyses and retraining closed-source models. The authors provided additional details about the curation process for MIMIC-1V3. However, some aspects, such as the evaluation of the correctness of the split and handling of missing sections, remain insufficiently detailed. The reviewers' final feedback was mixed. Reviewer Y732 and b8jm raised their scores, citing resolved concerns and strong responses, while XTQR and t2eU maintained their original ratings, citing unresolved issues with dataset analysis, insufficient comparisons, weak experimental support, and limited technical novelty.

Reviewer XTQR's concerns evolved but remained largely unresolved throughout the rebuttal process. Initially, XTQR sought a detailed pipeline error analysis to assess how inaccuracies in indications or findings affect downstream outputs and recommended introducing noise to indications to evaluate robustness. XTQR also requested statistical analyses to validate the correlation between indications, findings, and impressions and questioned the lack of comparisons with specialized radiology report generation methods. While the authors emphasized their method’s alignment with medical literature and their focus on dataset standardization, XTQR found this unconvincing. In the final feedback, XTQR reiterated the absence of sufficient analysis of the dataset, the lack of robust experimental validation, and inadequate comparisons with state-of-the-art methods, maintaining their original rating and intensifying the concerns.

---

### Decision · Program_Chairs · 2025-01-22

Reject